# Implicit Neural Representations for Joint Sparse-View CT Reconstruction

## Abstract

Computed Tomography (CT) plays a crucial role in both medical diagnostics and industrial quality control. Sparse-view CT, in particular, has advantages over standard CT for its reduced ionizing radiation but poses challenges due to its inherently ill-posed nature arising from undersampled measurement data. Implicit Neural Representations (INRs) have emerged as a promising solution, demonstrating effectiveness in sparse-view CT reconstruction. Given that modern CT often scans similar subjects, we propose to improve reconstruction quality via joint reconstruction of multiple objects using INRs. This approach can potentially leverage both the strengths of INRs and the statistical regularities across multiple objects. While existing techniques of INR joint reconstruction focus on enhancing convergence rates through meta-initialization, they do not optimize for final reconstruction quality. To fill this gap, we introduce a novel INR-based Bayesian framework that incorporates latent variables to capture inter-object relationships. These latent variables act as a continuously updated reference during the optimization process, thereby enhancing the quality of individual reconstructions. We conduct extensive experiments to evaluate various aspects such as reconstruction quality, susceptibility to overfitting, and generalizability. Our results demonstrate significant improvements over baselines in common numerical metrics, suggesting a step forward in CT reconstruction techniques. Our code will be released.

## 1 Introduction

Computed Tomography (CT) serves as a crucial non-invasive imaging tool in both medical diagnosis and industrial quality control. In CT, a series of X-ray projection images are captured from various angles to reconstruct an object's internal structure, solving an inverse problem. In specific situations, limiting the number of CT measurements can offer benefits such as reduced radiation exposure and cost management, which may lead to the use of sparse data. This sparsity complicates the reconstruction process, making it an ill-posed inverse problem. Such challenges arise not only in CT reconstruction but also across diverse computational tasks. Hence, while our study centers on sparse-view CT reconstruction, the core ideas are transferable to numerous inverse problems.

Various strategies tackle this challenge by incorporating auxiliary information. While many approaches learn the mapping from sparse-view to dense-view images using supervised learning (Zhang et al., 2018; Han & Ye, 2018; Zhu et al., 2018; Wu et al., 2021) or learn the image distribution solely from dense-view images (Song et al., 2022), they often necessitate extensive, domain-specific datasets which are difficult to obtain in practice. There are also works that adopt heuristic image priors, e.g. Total Variation (TV) (Sidky & Pan, 2008; Liu et al., 2013; Zang et al., 2018), or dense view images as priors (Chen et al., 2008; Shen et al., 2022) to assist in the reconstruction. They often lack domain-specific enhancements or require information from dense-view images. On a different tangent, many works explore the potential of implicit neural representations (INRs). Thanks to the continuous representation nature of INRs, these methods have consistently delivered promising results with limited data (Zang et al., 2021; Zha et al., 2022; Rückert et al., 2022; Wu et al., 2023).

Given INRs' proven capabilities in CT reconstruction and the known advantages of leveraging auxiliary information, we try to merge these two paradigms. Modern CT machines routinely scan similar subjects, such as patients in hospitals or analogous industrial products. This observation motivates us to investigate a novel question in this work:

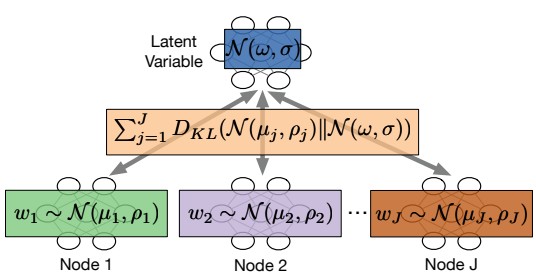

Figure 1: Framework of our proposed method. It uses latent variables to capture the relation among all reconstruction nodes. The latent variables are updated based on all nodes and regularizes each individual reconstruction via minimizing the KL divergence terms. Here, $w_j$ denotes the parameters of j-th node, distributed according to $\mathcal{N}(\boldsymbol{\mu}_j, \boldsymbol{\rho}_j)$, while $\mathcal{N}(\boldsymbol{\omega}, \boldsymbol{\sigma})$ specifies the prior distribution of parameters $\boldsymbol{w}_{1:J}$.

*Can INRs use the statistical regularities shared among different objects with similar representation to improve reconstruction quality through joint reconstruction?*

In our exploration of this research avenue, we found that several existing methods can be adapted for our purpose (Zhang et al., 2013; Ye et al., 2019; Tancik et al., 2021; Martin-Brualla et al., 2021; Kundu et al., 2022). Some previous works have exploited the statistical regularities among different objects borne in the INR networks' weights, but target different problems such as convergence rate (Tancik et al., 2021; Lee et al., 2021). A common practice in these methods is to find a network initialization that outperforms random initialization. However, these approaches may not fully capitalize on the available statistical regularities, as such information could be lost during the adaptation phase of the individual reconstructions.

To address our research question, we introduce a novel INR-based Bayesian framework designed to adaptively integrate prior information related to network weights throughout the training process. Specifically, we employ latent variables that capture common trend among different objects' neural representations, and subsequently apply this prior information to improve the accuracy of individual reconstructions. Both of these objectives are achieved by minimizing the Kullback-Leibler (KL) divergence between the prior and the approximated posterior distributions associated with the neural representation networks. Importantly, our framework can automatically adjust the regularization effect of the prior information based on the similarity among neural representation networks, allowing for a broader range of applications in reconstructing diverse images. Overall, our framework provides a robust solution to the challenges posed by sparse data and varied reconstructions in CT imaging. An illustration of our proposed method is provided in Figure 1.

**Our Contributions:** i) We explore a novel problem of INR-based joint reconstruction in the context of CT imaging, supported by a comprehensive review of existing methods that could be adapted to address this challenge. ii) We propose a principled Bayesian framework that adaptively integrates prior information throughout the training process for enhanced reconstruction quality. iii) Through extensive experiments, we evaluate various facets of reconstruction performance using common numerical metrics. Our results establish that our method either outperforms or is competitive with existing INR-based baselines, suggesting notable advancements in the field of CT reconstruction.

## 2 RELATED WORKS

We briefly outline key studies related to our focal areas, with a comprehensive understanding of NeRF and INR available in the survey (Tewari et al., 2022).

**Neural Radiance Fields.** Coordinates-based Multi-Layer Perceptrons (MLPs) have transitioned from traditional discrete representations to implicit neural representations (INRs) by addressing high-frequency function detailing issues (Jacot et al., 2018; Tancik et al., 2020). Neural Radiance Fields (NeRF), a state-of-the-art INR approach, models continuous scenes using spatial coordinates and viewing angles, incorporating transmittance effects during ray-tracing (Mildenhall et al., 2021; Barron et al., 2021; 2022). Specifically, NeRF-wild (Martin-Brualla et al., 2021) differentiated between static and transient scene aspects, an approach echoed in video representations (Li et al., 2021; Mai & Liu, 2022).

**INR for CT Reconstruction.** INRs' potential in CT reconstruction has been exploited in various ways. While Sun et al. (2021) focused on representing sparse measurements, Zang et al. (2021) combined INRs with total variation and non-local priors for CT reconstruction. Notable advancements include cone-beam CT optimization (Zha et al., 2022) and adaptive hierarchical octree representation (Rückert et al., 2022). Wu et al. (2023) also improved reconstruction precision using reprojections on inferred density fields.

Building on the groundwork laid by INR-based approaches, several techniques have emerged to leverage prior information in joint CT reconstruction. Meta-learning's application in CT reconstruction was first introduced by Tancik et al. (2021), using techniques like MAML (Nichol et al., 2018). Later, Lee et al. (2021) introduced sparsity to the initialization, while Chen & Wang (2022) proposed using transformers. For scene representation, Kundu et al. (2022) applied federated learning to obtain the prior information. Lastly, while other INR-based CT reconstructions like Shen et al. (2022) use priors from pre-reconstructed images, and Reed et al. (2021) rely on finding a template image from 4DCT, their practical limitations led to their exclusion from our comparative analysis.

## 3   PROBLEM STATEMENT AND PRELIMINARIES

Mathematically, the CT acquisition process can be formulated as a linear equation: $\boldsymbol{y} = \boldsymbol{A}\boldsymbol{x} + \boldsymbol{\epsilon}$, where $\boldsymbol{x} \in \mathbb{R}^m$ represents the unknown object of interest and $\boldsymbol{y} \in \mathbb{R}^n$ symbolizes the noisy measurements. These measurements arise from the interaction between the measurement matrix $\boldsymbol{A} \in \mathbb{R}^{n \times m}$ and the object, with $\boldsymbol{\epsilon} \in \mathbb{R}^n$ accounting for the associated measurement noise. The task in CT is to infer the unknown object $\boldsymbol{x}$ from the acquired CT measurements $\boldsymbol{y}$. The inherent challenge lies in the common sparsity of these measurements, resulting in $m > n$. This makes the reconstruction problem ill-posed.

The INR designed for CT reconstruction is a function $f_{\boldsymbol{w}} : \mathbb{R}^3 \to \mathbb{R}^1$ parameterized by $\boldsymbol{w}$. It maps the spatial coordinates of the object to its intensity in a continuous three-dimensional space. INR consists of two components, formulated as $f_{\boldsymbol{w}} = \mathcal{M} \circ \gamma$. Here, $\gamma : \mathbb{R}^3 \to \mathbb{R}^d$ servers as the position encoding (Tancik et al., 2020; Barron et al., 2022; Martel et al., 2021; Müller et al., 2022), while $\mathcal{M} : \mathbb{R}^d \to \mathbb{R}^1$ acts as the neural representation. Typically, $\mathcal{M}$ is a multi-layer perceptron (MLP). The function $f_{\boldsymbol{w}}(\cdot)$ takes a coordinate $\boldsymbol{c}_i \in \mathbb{R}^3$ and maps it to the intensity value $v \in \mathbb{R}^1$. For a full set of coordinate $\boldsymbol{C} := \{\boldsymbol{c}_1, \boldsymbol{c}_2, \dots, \boldsymbol{c}_N\}$, the INR outputs the representation of the entire object as $\mathcal{F}_{\boldsymbol{w}}(\boldsymbol{C}) := \{f_{\boldsymbol{w}}(\boldsymbol{c}_1), f_{\boldsymbol{w}}(\boldsymbol{c}_2), \dots, f_{\boldsymbol{w}}(\boldsymbol{c}_N)\}$. The optimization procedure for INR-based reconstruction involves minimizing the loss function: $\ell(\boldsymbol{w}) := \|\boldsymbol{A}\mathcal{F}_{\boldsymbol{w}}(\boldsymbol{C}) - \boldsymbol{y}\|_2^2$.

**Joint Reconstruction Problem.** We aim to simultaneously recover $J$ objects $\boldsymbol{x}_{1:J}$ using their corresponding measurements $\boldsymbol{y}_{1:J}$ and measurement matrices $\boldsymbol{A}_{1:J}$. The joint reconstruction problem can be mathematically formulated as:

$$\boldsymbol{w}_{1:J}^* = \arg\min_{\boldsymbol{w}_{1:J}} \sum_{j=1}^{J} \ell_j(\boldsymbol{w}_j), \quad \ell_j(\boldsymbol{w}_j) := \|\boldsymbol{A}_j \mathcal{F}_{\boldsymbol{w}_j}(\boldsymbol{C}) - \boldsymbol{y}_j\|_2^2. \tag{1}$$

We believe that by introducing a dynamic prior that not only links all the models $\boldsymbol{w}_{1:J}$ during training but also updates in response to their optimization, a Bayesian framework can provide a principled way to exploit the shared statistical regularities among different objects, thereby enhancing the quality of joint reconstruction.

### 3.1   EXISTING METHODS AVAILABLE FOR JOINT RECONSTRUCTION

Although several existing methods are originally designed for different problems and do not employ a Bayesian framework, they also align well with the objective highlighted in Equation (1). In the following sections, we delve into these methods in greater detail. Empirical evaluations suggest that some of these techniques can outperform the individual reconstruction approach, as discussed in Section 5. Thus, we also benchmark these methods against our proposed Bayesian framework.

**Composite of Static and Transient Representations.** Martin-Brualla et al. (2021) introduce a composite representation approach, known as NeRFWild, designed to manage variable illumination and transient occluders in a collective of observations. While CT does not involve variable illumination, their concept of combining "static" and "transient" components can be adapted for our context, which we term `INRWild`.

Let $\mathcal{G}_{\phi}$ represent the neural representation for the static component and $\mathcal{H}_{\boldsymbol{w}}$ signify the transient component. For a given set of $J$ objects, each object-associated reconstruction node has its distinct transient network $\boldsymbol{w}_j$ and corresponding transient feature $\boldsymbol{b}_j$. In contrast, the static network $\phi$ is shared across all nodes. The objective for this framework is formulated as:

$$\phi^*, \boldsymbol{w}_{1:J}^*, \boldsymbol{b}_{1:J}^* = \underset{\phi, \boldsymbol{w}_{1:J}, \boldsymbol{b}_{1:J}}{\arg\min} \sum_{j=1}^{J} \|\boldsymbol{A}_j \left( \mathcal{H}_{\boldsymbol{w}_j} \left( \boldsymbol{b}_j, \mathcal{G}_{\phi}^{\backslash r}(\boldsymbol{C}) \right) + \mathcal{G}_{\phi}^r(\boldsymbol{C}) \right) - \boldsymbol{y}_j\|_2^2. \tag{2}$$

Here, $\mathcal{G}_{\phi}^r(\boldsymbol{C})$ represents the static intensity, and $\mathcal{G}_{\phi}^{\backslash r}(\boldsymbol{C})$ serves as intermediate features for the transient network. For a more detailed explanation and a schematic depiction of this framework, readers can refer to the Appendix C.1. At its core, `INRWild` emphasizes training the static network $\phi$, which embodies most learnable parameters, using aggregated losses. Concurrently, the individual parameters, characterized by $\boldsymbol{w}_{1:J}$ and $\boldsymbol{b}_{1:J}$, are refined based on $\phi$'s characteristics.

**Model-agnostic Meta-learning (MAML)**: Meta-learning aims to train a network in a way that it can quickly adapt to new tasks (Nichol et al., 2018; Fallah et al., 2020). Several INR-based works have employed `MAML` to obtain a meta-learned initialization, thereby accelerating the convergence or enabling model compression (Tancik et al., 2021; Lee et al., 2021). In the `MAML` framework, computational cycles are organized into "inner loops" and "outer loops", indexed by $k = 1 \dots K$ and $t = 1, \dots, T$ respectively. For each node $j = 1, \dots, J$, the networks $\boldsymbol{w}_{1:J}$ are initialized according to the meta neural representation $\boldsymbol{w}_{1:J}^{(0)} = \boldsymbol{\theta}$. These networks then undergo $K$ steps of inner-loop learning: $\boldsymbol{w}_j^{(k)} = \boldsymbol{w}_j^{(k-1)} - \eta \nabla_{\boldsymbol{w}_j} \ell_j(\boldsymbol{w}_j^{(k-1)})$, where $\eta$ is the inner-loop learning rate. After these $K$ steps, the meta network $\boldsymbol{\theta}$ updates as follows:

$$\boldsymbol{\theta}^t = \boldsymbol{\theta}^{t-1} - \alpha \frac{1}{J} \sum_{j=1}^{J} \nabla_{\boldsymbol{\theta}} \ell_j(\boldsymbol{w}_j^{(K)}), \quad \nabla_{\boldsymbol{\theta}} \ell_j(\boldsymbol{w}_j^{(K)}) = \nabla_{\boldsymbol{w}_j} \ell_j(\boldsymbol{w}_j^{(K)}) \frac{\partial \boldsymbol{w}_j^{(K)}}{\partial \boldsymbol{\theta}}, \tag{3}$$

where $\alpha$ is the outer-loop learning rate. After $T$ steps of outer-loop optimization, the meta-learned neural representation $\boldsymbol{\theta}^T$ serves as an effective initialization for individual reconstructions.

**Federated Averaging (FedAvg)**: Kundu et al. (2022) suggested to employ `FedAvg` (McMahan et al., 2017) as the optimization framework of the meta-learned initialization. Like `MAML`, `FedAvg` also consists of inner and outer loops. The inner loop is identical to `MAML`. Whereas, the outer loop simplifies the meta network optimization by averaging all individual networks, represented as $\boldsymbol{\theta} = \frac{1}{J} \sum_j \boldsymbol{w}_j^{(K)}$. Essentially, the meta network acts as the centroid of all networks. [1]

## 4  A NOVEL BAYESIAN FRAMEWORK FOR JOINT RECONSTRUCTION

In this section, we introduce `INR-Bayes`, our Bayesian framework for INR joint reconstruction.

**Motivation.** A method that uses a composition of static and transient components operates under the assumption that all the representations substantially overlap. This may be true in 3D scene reconstruction, where observations are taken from different viewpoints of the same object. Our empirical findings on `INRWild` indicate such methods do not work efficiently in CT reconstruction and other image-level reconstruction tasks. Meta-learned initialization methods train a meta-model to capture a conceptual common representation, which can then be flexibly adapted to individual objects. However, such methods subsequently adapt the models purely based on local measurements, making them prone to the notorious overfitting issue in iterative methods of CT reconstruction (Herman & Odhner, 1991; Effland et al., 2020), as demonstrated in Section 5.

By considering the meta-model, which we denote by $\boldsymbol{\omega}$ in the sequel, a latent variable that updates based on individual networks and uses it as a reference for individual training, a Bayesian framework provides a principled way to conduct this process.

**Definition and Notation.** We introduce distribution to the networks $\boldsymbol{w}_{1:J}$ for $J$ objects, and define latent variables $\{\boldsymbol{\omega}, \boldsymbol{\sigma}\}$ that parameterize an *axis-aligned multivariate* Gaussian prior $\mathcal{N}(\boldsymbol{\omega}, \boldsymbol{\sigma})$

---

[1] It is noteworthy that FedAvg can also be regarded as using a specific first-order algorithm of MAML called Reptile (Nichol et al., 2018) and setting the outer-loop learning rate to 1.

from which the weights are generated. These latent variables collectively serve to capture the shared trends within the network, effectively quantifying the mutual information across different objects. To simplify the model, we assume the *conditional independence* among all objects: $p(\boldsymbol{w}_{1:J}|\boldsymbol{\omega}, \boldsymbol{\sigma}) = \prod_{j=1}^{J} p(\boldsymbol{w}_j|\boldsymbol{\omega}, \boldsymbol{\sigma})$. This assumption of conditional independence allows us to decompose the variational inference into a separable optimization problem, thereby facilitating more efficient parallel computing.

Given that the measurements of the objects $\boldsymbol{y}_1, \dots, \boldsymbol{y}_J$ are mutually independent and that each network focuses on a specific object, the posterior distribution of network weights and latent variables can be derived using the Bayes' rule as $p(\boldsymbol{w}_{1:J}, \boldsymbol{\omega}, \boldsymbol{\sigma}|\boldsymbol{y}_{1:J}) \propto p(\boldsymbol{\omega}, \boldsymbol{\sigma}) \prod_{j=1}^{J} p(\boldsymbol{y}_j|\boldsymbol{w}_j)p(\boldsymbol{w}_j|\boldsymbol{\omega}, \boldsymbol{\sigma})$. While this posterior enables various forms of deductive reasoning, inferring the true posterior is often computationally challenging or intractable. Moreover, the selection of an appropriate prior $p(\boldsymbol{\omega}, \boldsymbol{\sigma})$ poses its own difficulties (Wenzel et al., 2020; Fortuin et al., 2022). To tackle these issues, we present an algorithm that aims at maximizing the marginal likelihood $p(\boldsymbol{y}_{1:J}|\boldsymbol{\omega}, \boldsymbol{\sigma})$ in the sequel. The details of derivations are provided in Appendix B.

## 4.1 OPTIMIZATION METHOD

To optimize the marginal likelihood $p(\boldsymbol{y}_{1:J}|\boldsymbol{\omega}, \boldsymbol{\sigma})$, we approximate the posterior distribution of the network weights $\boldsymbol{w}_{1:J}$ using variational inference techniques (Kingma & Welling, 2013; Blei et al., 2017). Specifically, we introduce the *factorized* variational approximation $q(\boldsymbol{w}_{1:J}) = \prod_{j=1}^{J} q(\boldsymbol{w}_j)$, employing an *axis-aligned multivariate* Gaussian for the variational family, i.e. $q(\boldsymbol{w}_j) = \mathcal{N}(\boldsymbol{\mu}_j, \boldsymbol{\rho}_j)$.

**Variational Expectation Maximization.** To maximize the marginal likelihood, we use the evidence lower bound (ELBO):

$$ELBO\left(q(\boldsymbol{w}_{1:J}), \boldsymbol{\omega}, \boldsymbol{\sigma}\right) = \mathbb{E}_{q(\boldsymbol{w}_{1:J})} \log \frac{p(\boldsymbol{y}_{1:J}, \boldsymbol{w}_{1:J}|\boldsymbol{\omega}, \boldsymbol{\sigma})}{q(\boldsymbol{w}_{1:J})}. \tag{4}$$

The ELBO is optimized using Expectation Maximization (EM) (Dempster et al., 1977), a two-stage iterative algorithm involving an E-step and an M-step. Generally, each EM cycle improves the marginal likelihood $p(\boldsymbol{y}_{1:J}|\boldsymbol{\omega}, \boldsymbol{\sigma})$ unless it reaches a local maximum.

**E-step.** At this stage, the latent variables $\{\boldsymbol{\omega}, \boldsymbol{\sigma}\}$ are held fixed. The aim is to maximize ELBO by optimizing the variational approximations $q(\boldsymbol{w}_{1:J})$. By assumption, the objective can be separately optimized for each network. Specifically, each network minimizes:

$$\mathcal{L}\left(q(\boldsymbol{w}_j)\right) = -\mathbb{E}_{q(\boldsymbol{w}_j)} \log p(\boldsymbol{y}_j|\boldsymbol{w}_j) + D_{KL}(q(\boldsymbol{w}_j)\|p(\boldsymbol{w}_j|\boldsymbol{\omega}, \boldsymbol{\sigma})). \tag{5}$$

The minimization of the negative log-likelihood term is achieved through the minimization of the squared error loss of reconstruction (see Equation (1)). The KL divergence serves as a regularization constraint on the network weights, pushing $\boldsymbol{w}_j$ to be closely aligned with a conditional prior determined by $\{\boldsymbol{\omega}, \boldsymbol{\sigma}\}$. These parameters represent the collective mean and variance of all the networks in the ensemble. *The KL divergence thus serves to couple the neural representations across networks, allowing them to inform each other.*

**M-step.** After obtaining the optimized variational approximations $q(\boldsymbol{w}_{1:J})$, we proceed to maximize the ELBO with respect to the latent variables $\{\boldsymbol{\omega}, \boldsymbol{\sigma}\}$:

$$ELBO(\boldsymbol{\omega}, \boldsymbol{\sigma}) \propto \sum_{j=1}^{J} \mathbb{E}_{q(\boldsymbol{w}_j)} \log p(\boldsymbol{w}_j|\boldsymbol{\omega}, \boldsymbol{\sigma}). \tag{6}$$

Equation (6) allows for a closed-form solution of $\{\boldsymbol{\omega}, \boldsymbol{\sigma}\}$, derived by setting the derivative of the ELBO to zero:

$$\boldsymbol{\omega}^* = \frac{1}{J} \sum_{j=1}^{J} \boldsymbol{\mu}_j, \quad \boldsymbol{\sigma}^* = \frac{1}{J} \sum_{j=1}^{J} \boldsymbol{\rho}_j + (\boldsymbol{\mu}_j - \boldsymbol{\omega}^*)^2. \tag{7}$$

In our framework, $\boldsymbol{\omega}$ serves as a collective mean of individual network weights, while $\boldsymbol{\sigma}$ provides an adaptive measure of dispersion, factoring in both individual variances and deviations from the collective mean. We note the KL divergence term, introduced in the preceding E-step objective (see Equation (5)), operates element-wise. *During the training process, weight elements with larger values of $\boldsymbol{\sigma}$ are less regularized, thereby offering a flexible, self-adjusting regularization scheme that pushes all weights toward the latent mean $\boldsymbol{\omega}$.*

---

**Algorithm 1** INR-Bayes: Joint reconstruction of INR using Bayesian framework

---

    **Input:** $\boldsymbol{\mu}_{1:J}^{(0,0)}, \boldsymbol{\pi}_{1:J}^{(0,0)}, \boldsymbol{\omega}^0, \boldsymbol{\sigma}^0, \eta, \beta, T, R$

    **Output:** $\boldsymbol{\mu}_{1:J}^{(R,T)}, \boldsymbol{\pi}_{1:J}^{(R,T)}, \boldsymbol{\omega}^R, \boldsymbol{\sigma}^R$

1: **for** $r = 1$ to $R$ **do**
2:     **for** $j = 1, \ldots, J$ **in parallel do**
3:       $\lfloor$   **NodeUpdate**$(\boldsymbol{\omega}^{r-1}, \boldsymbol{\sigma}^{r-1})$
4:      After the E-step of each network, collecting $\boldsymbol{\mu}_{1:J}^{(r,T)}, \boldsymbol{\pi}_{1:J}^{(r,T)}$.
5:      ▷ *Compute the optimal latent variable $\omega, \sigma$.*                                ◁
6:      $\boldsymbol{\omega}^r = \frac{1}{J} \sum_{j=1}^{J} \boldsymbol{\mu}_j^{(r,T)}$
7:      $\boldsymbol{\sigma}^r = \frac{1}{J} \sum_{j=1}^{J} \log\left(1 + \exp(\boldsymbol{\pi}_j^{(r,T)})\right) + (\boldsymbol{\mu}_j^{(r,T)} - \boldsymbol{\omega}^r)^2$

8: **NodeUpdate**$(\boldsymbol{\omega}^r, \boldsymbol{\sigma}^r)$:
9: **for** $t = 1, \ldots, T$ **in parallel do**
10:     ▷ *Sample $\widehat{\boldsymbol{w}}_j$.*                                                            ◁
11:     $\widehat{\boldsymbol{w}}_j^t \sim \boldsymbol{\mu}_j^{(r,t)} + \log\left(1 + \exp(\boldsymbol{\pi}_j^{(r,t)})\right)\mathcal{N}(\boldsymbol{0}, \boldsymbol{I})$
12:     ▷ *Compute the loss function.*                            ◁
13:     $\mathcal{L}(\boldsymbol{\mu}_j, \boldsymbol{\rho}_j) = \|\boldsymbol{A}_j(\mathcal{F}_{\widehat{\boldsymbol{w}}_j^t}(\boldsymbol{C})) - \boldsymbol{y}_j\|_2^2 + \beta D_{KL}(q(\boldsymbol{w}_j)\|p(\boldsymbol{w}_j|\boldsymbol{\omega}^r, \boldsymbol{\sigma}^r))$
14:     ▷ *SGD on the variational approximation $\boldsymbol{\mu}_j, \boldsymbol{\rho}_j$ with learning rate $\eta$.*    ◁
15:     $\boldsymbol{\mu}_j^{(r,t+1)} = \boldsymbol{\mu}_j^{(r,t)} - \eta\frac{\partial\mathcal{L}}{\boldsymbol{\mu}_j}, \boldsymbol{\pi}_j^{(r,t+1)} = \boldsymbol{\pi}_j^{(r,t)} - \eta\frac{\partial\mathcal{L}}{\boldsymbol{\pi}_j}$

---

### 4.2 IMPLEMENTATION

We delve into the intricacies of implementation, addressing in particular the computational challenges associated with Equation (5). A summary of our method can be found in Algorithm 1.

**Variational Approximation.** Given that the expected likelihood in Equation (5) is generally intractable, we resort to Monte Carlo (MC) sampling to provide an effective estimation. Moreover, we introduce an additional hyperparameter $\beta$ for the KL divergence to balance the trade-off between model complexity and overfitting. Linking the likelihood with the square error loss, for any node $j$, the effective loss function can be expressed as:

$$\mathcal{L}\left(q(\boldsymbol{w}_j)\right) \approx \|\boldsymbol{A}_j(\mathcal{F}_{\widehat{\boldsymbol{w}}_j}(\boldsymbol{C})) - \boldsymbol{y}_j\|_2^2 + \beta D_{KL}(q(\boldsymbol{w}_j)\|p(\boldsymbol{w}_j|\boldsymbol{\omega}, \boldsymbol{\sigma})), \tag{8}$$

where $\widehat{\boldsymbol{w}}_j$ denotes a sample from $q(\boldsymbol{w}_j)$. We only do MC sampling once at each iteration, which works efficiently in practice.

**Reparameterization Tricks.** To facilitate the gradient-based optimization schemes, we utilize the reparameterization trick (Kingma & Welling, 2013):

$$q(\boldsymbol{w}_j) = \boldsymbol{\mu}_j + \log\left(1 + \exp\left(\boldsymbol{\pi}_j\right)\right)\mathcal{N}(\boldsymbol{0}, \boldsymbol{I}). \tag{9}$$

Here, we additionally deploy the softplus function in parameterizing the variance $\boldsymbol{\sigma}_j$ with the variable $\boldsymbol{\pi}_j$ to ensure the non-negativity of the variance of the variational approximation.

The EM algorithm operates through alternating E and M steps. In the E-step, we perform $T$ iterations to achieve the locally optimal variational approximations. Following this, the M-step utilizes the closed-form solution (see Equation (7)) to achieve an efficient parameter update. The entire cycle is executed for $R$ rounds to ensure convergence. Finally, the parameters $\boldsymbol{\mu}_{1:J}$ serve as the weights for individual neural representation, while $\boldsymbol{\omega}$ is used as the weights for meta neural representation.

## 5 EXPERIMENTS

**Dataset.** Our study utilizes three CT datasets: 4DCT on the lung area (Castillo et al., 2009), LungCT from the Medical Segmentation Decathlon (Antonelli et al., 2022), and BrainCT from the Brain CT Hemorrhage Challenge (Flanders et al., 2020). Additionally, we include a natural image dataset CelebA (Liu et al., 2015) to evaluate the generalizability to broader applications.

| Experiment | Metrics | FBP | SIRT | SingleINR | INRWild | FedAvg | MAML | INR-Bayes |
|---|---|---|---|---|---|---|---|---|
| Intra-patient | PSNR | 26.50 ±0.06 | 28.81 ±0.06 | 32.80 ±0.11 | 28.46 ±0.07 | 32.42 ±0.08 | 33.26 ±0.10 | **33.90** ±0.10 |
| | SSIM | 0.568 ±0.002 | 0.719 ±0.002 | 0.815 ±0.002 | 0.674 ±0.003 | 0.808 ±0.002 | 0.825 ±0.002 | **0.840** ±0.002 |
| Inter-patient Lung | PSNR | 24.97 ±0.14 | 28.32 ±0.13 | 32.64 ±0.27 | 25.05 ±0.18 | 31.68 ±0.19 | 33.13 ±0.22 | **33.75** ±0.20 |
| | SSIM | 0.503 ±0.004 | 0.678 ±0.005 | 0.821 ±0.007 | 0.560 ±0.08 | 0.807 ±0.006 | 0.833 ±0.006 | **0.847** ±0.006 |
| Inter-patient Brain | PSNR | 17.68 ±0.34 | 20.74 ±0.49 | 25.56 ±1.14 | 21.81 ±0.37 | 24.46 ±0.96 | **25.81** ±1.14 | **25.84** ±1.15 |
| | SSIM | 0.398 ±0.004 | 0.498 ±0.004 | 0.801 ±0.017 | 0.638 ±0.09 | 0.761 ±0.011 | **0.823** ±0.013 | **0.823** ±0.014 |
| 4DCT | PSNR | 25.19 ±0.03 | 28.61 ±0.03 | 34.07 ±0.04 | 33.76 ±0.04 | 34.67 ±0.04 | 34.35 ±0.05 | **34.79** ±0.04 |
| | SSIM | 0.529 ±0.001 | 0.746 ±0.001 | 0.877 ±0.001 | 0.863 ±0.001 | 0.885 ±0.001 | 0.881 ±0.001 | **0.894** ±0.001 |

Table 1: Results from intra-patient, inter-patient joint reconstruction, and joint reconstruction across temporal phases in 4DCT. The highest average PSNR/SSIM values that are statistically significant are bolded. Our method, INR-Bayes, consistently achieves the best performance across datasets.

**Comparison Methods.** We compare our approach with the following methods: i) Classical techniques: Filtered Back Projection (`FBP`) and Simultaneous Iterative Reconstruction Technique (`SIRT`); ii) Naive INR-based single reconstruction method, denoted as `SingleINR`; iii) `FedAvg`, a federated averaging approach proposed by Kundu et al. (2022); iv) `MAML`, A meta-learning technique as discussed by Tancik et al. (2021); v) `INRWild`, a method adapted from NeRFWild (Martin-Brualla et al., 2021). `FBP` and `SIRT` are classical methods that do not use neural networks, while all other methods employ an identical INR network as described in the next paragraph.

**INR Network Configuration.** The same backbone and associated configurations are applied to ensure a fair comparison. All INR-based methods employ the SIREN architecture (Sitzmann et al., 2020) coupled with the same positional embedding (Tancik et al., 2020). In alignment with the `INRWild` design, we utilize an 8-layer SIREN network for the static segment and a 4-layer SIREN for each transient component. Additional details are available in the Appendices C.2 and C.3.

**CT Configuration.** We simulate CT projections using the Tomosipo package (Hendriksen et al., 2021) with a parallel beam. Experiments on 4DCT and BrainCT use projections from 40 angles across $180°$, while others use 60 angles. Practical applicability is further tested under a 3D cone-beam CT setting, detailed in Appendix E.1.

**Experiment Configurations.** i) Intra-patient: 10 equidistant slices from a patient's lung center. ii) Inter-patient: 10 slices from different patients, each from a similar upper-body/head position. iii) 4DCT: 10 temporal phases from one 4DCT slice.

**Metrics.** We primarily evaluate using Peak Signal to Noise Ratio (PSNR) and Structural Similarity Index Measure (SSIM), with metrics referenced against ground-truth images. We calulate mean and standard error over all reconstructioned images in each experiment.

## 5.1 RESULTS

**Reconstruction Performance.** Table 1 presents the average metrics across various datasets. Our method consistently achieves the top average PSNR/SSIM values, underscoring its proficiency in exploiting inherent trends across slices. The superiority of our approach is more pronounced when the images demonstrate an inherent transition pattern, as observed in both inter-patient and intra-patient experiments. Meta-learning mostly ranks as the second-best joint reconstruction method, with an exception in the 4DCT dataset where pronounced image similarities exist. This highlights the advantages of utilizing averaging as a prior under such circumstances.

The visual comparisons in Figure 2 and Figure 3 further substantiate our findings. The reconstruction of `SingleINR` shows noticeable artifacts. Although `FedAvg` and `MAML` achieve higher PSNR and SSIM due to smoother reconstructions, they also sacrifice some image details. In contrast, our `INR-Bayes` method consistently delivers superior visual quality, balancing smoothness and detail.

**Comparison with Different Numbers of Nodes and Angles.** Figure 4a demonstrates that all methods see an improvement in average PSNR values as the number of scanning angles increases. Methods that leverage prior information, such as `FedAvg`, `MAML`, and ours `INR-Bayes` outperform `singleINR` when the number of angles is limited. With only 20 angles, `FedAvg`'s performance is

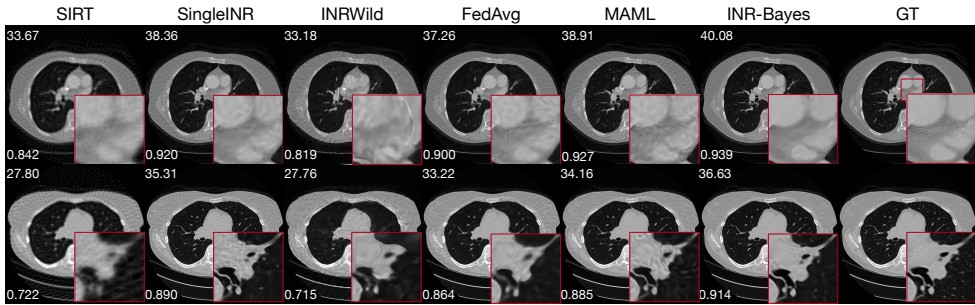

Figure 2: Visual comparison for intra-patient joint reconstruction. Enlarged areas are highlighted in red insets. PSNR values are on the top left, with SSIM values on the bottom left.

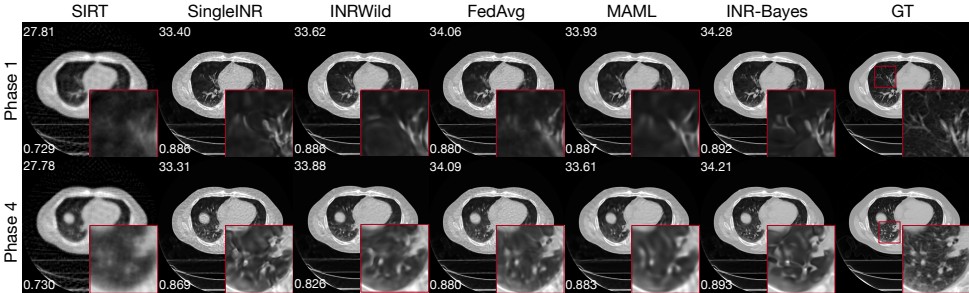

Figure 3: Visual comparison for joint reconstruction across 4DCT temporal phases. Enlarged areas are highlighted in red insets. PSNR values are on the top left, with SSIM values on the bottom left.

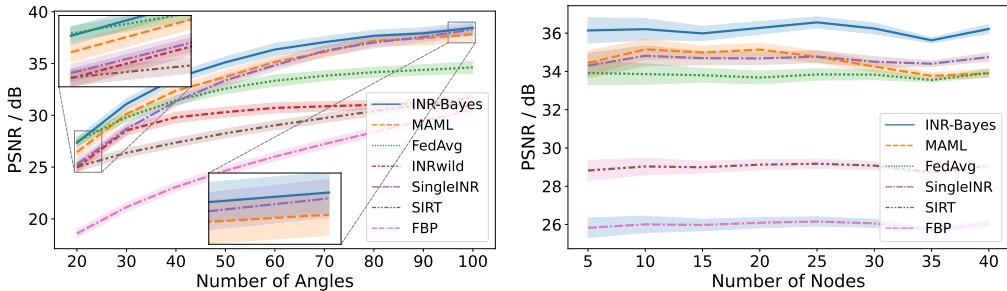

(a) Performance across different numbers of scanning angles.

(b) Different numbers of nodes. Individual reconstruction methods are presented as reference.

Figure 4: Results of the impact of varying scanning angles and nodes on intra-patient LungCT.

on par with our method, indicating that simple averaging can be effective in extremely data-scarce scenarios. However, as the number of angles grows, both ours `INR-Bayes` and `MAML` surpass FedAvg. Remarkably, our `INR-Bayes` method generally yields the best results. It is also worth noting that the performance gap between `singleINR` and ours `INR-Bayes` narrows as more data becomes available, suggesting that while the prior information is useful in sparse data situations, its advantage diminishes in the data-rich environment. In Figure 4b, our method consistently delivers superior performance compared to other methods across a range of node counts. `MAML` shows strong results when the node count is between 5 and 25, but experiences a decline in performance, eventually matching that of `FedAvg` when the node count reaches 40. This drop indicates that `MAML` might struggle to capture the shared features when many nodes are participating in the joint reconstruction.

**Overfitting.** Iterative reconstruction methods tend to overfit when applied to limited data (Herman & Odhner, 1991; Effland et al., 2020). In contrast, Bayesian frameworks have demonstrated robustness against overfitting (MacKay, 1992; Neal, 2012; Blundell et al., 2015). To validate this,

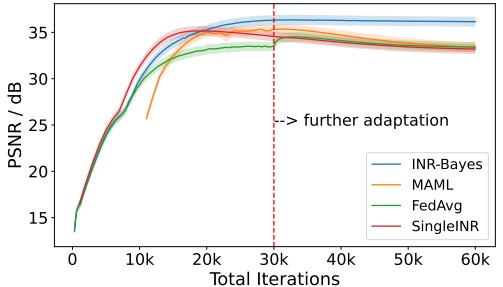 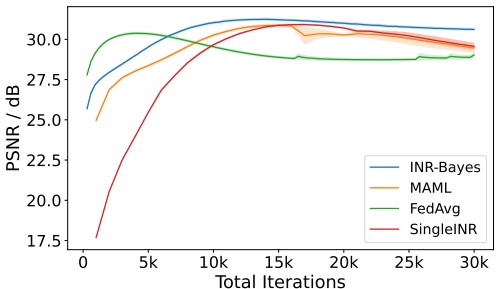

Figure 5: After the initial 30,000 iterations, each joint reconstruction method undergoes an additional 30,000 iterations for further adaptation.

Figure 6: Reconstruction of 5 new patients using the learned prior from 10 other patients, compared to individual reconstruction.

|      | SingleINR | FedAvg | MAML | INR-Bayes |
|------|-----------|--------|------|-----------|
| PSNR | $30.22 \pm_{0.12}$ | $29.89 \pm_{0.16}$ | $30.42 \pm_{0.14}$ | $\mathbf{31.31} \pm_{0.12}$ |
| SSIM | $0.769 \pm_{0.003}$ | $0.754 \pm_{0.004}$ | $0.770 \pm_{0.003}$ | $\mathbf{0.799} \pm_{0.003}$ |

Table 2: Adaptation on new patients using priors learned from other patients compared to individual reconstruction.

we extend the training iterations from 30K to 60K, designating the latter half as a pure adaptation phase. As shown in Figure 5 on inter-patient LungCT, the learning curves of baselines deteriorate in the long run, indicating overfitting on the measurement noise. Conversely, our approach maintains a consistent level of reconstruction quality once the optimal performance is achieved, underscoring the robustness of our framework. We note that determining an exact stopping criterion is challenging without reference ground truth, making such robustness highly valuable in practice.

**Applying to Unseen Data using Learned Prior.** We apply the acquired prior from the inter-patient experiment to guide the reconstruction of test subjects in the LungCT dataset. Specifically, we select 5 consecutive slices from new patients, choosing slices from the same anatomical location the prior has been trained. The prior information is solely utilized to guide the reconstruction and is not updated during the process. Table 2 shows that `FedAvg` fails to improve the reconstruction quality compared with `singleINR`, suggesting its learned meta neural representation struggles to generalize to unseen data. In contrast, both `MAML` and ours `INR-Bayes` effectively leverage their trained priors for improved reconstruction, with our method showing notably better metrics. Figure 6 presents performance curves of different methods. All joint reconstruction methods converge faster than individual reconstruction. Initially, `FedAvg` converges the fastest, but as training progresses, both `MAML` and `INR-Bayes` surpass it. Additionally, the results reconfirm the robustness of ours `INR-Bayes` against overfitting, a problem that other methods cannot avoid.

**Broader Application.** We also conduct experiments on the CelebA dataset to evaluate the generalizability of different methods to natural images. Results are relegated to Appendix E.2.

## 6 DICUSSION AND CONCLUSION

We introduced a novel INR-based Bayesian framework tailored for joint CT reconstruction in this study. Through extensive experiments, our method has effectively showcased its ability to leverage the statistical regularities inherent in the sparse measurement of multiple objects to improve individual reconstructions. This capability allows our approach to outperform competing methods in terms of reconstruction quality, robustness to overfitting as well as generalizability. While the primary focus of our method has been on joint CT reconstruction, its underlying principles hold potential applicability across a variety of inverse problems plagued by the challenges of sparse measurements.

**Limitation.** We recognize that INR-based methods outperform conventional ones but require more computation, making their efficiency a crucial focus for future research. Additionally, the metrics employed in our study may not always correlate with clinical evaluations (Renieblas et al., 2017; Verdun et al., 2015). If applied in a medical application, clinical verification of our method remains essential to understand its practical implications and efficacy in a given clinical setting.

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

## A ILLUSTRATIVE OVERVIEW OF OUR METHOD

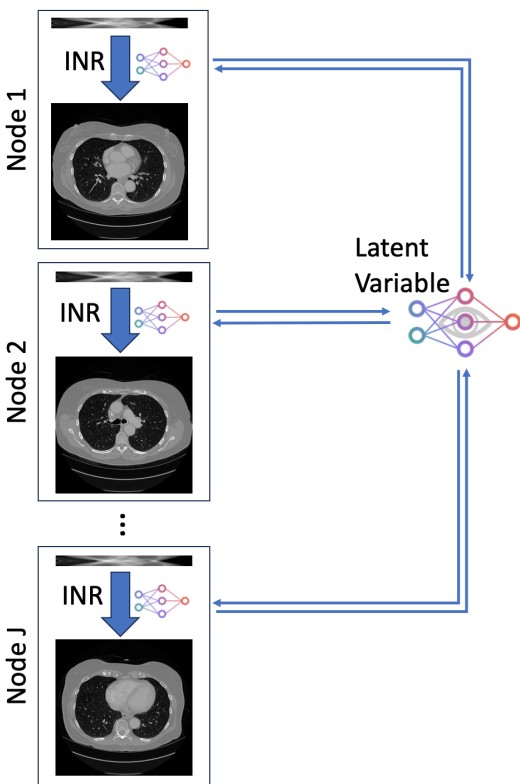

Figure 7: Overview of Our Proposed Method: This schematic represents our INR-based Bayesian framework for joint CT reconstruction. It features $J$ nodes collaboratively reconstructing images, with a central latent variable capturing the relationships between these nodes. The network-like structure of the latent variable and local parameters are depicted for illustrative purposes. They are in the form of axis-aligned Gaussian distributions, characterized by mean and variance components in our Bayesian framework.

## B VARIATIONAL EXPECTATION MAXIMIZATION OF THE MARGINAL LIKELIHOOD

To optimize the marginal likelihood $p(\boldsymbol{y}_{1:J}|\boldsymbol{\omega}, \boldsymbol{\sigma})$, we leverage a variational approximation of the network weights $q(\boldsymbol{w}_{1:J})$. We start by rewriting the marginal likelihood as follows:

$$\log p(\boldsymbol{y}_{1:J}|\boldsymbol{\omega}, \boldsymbol{\sigma}) = \mathbb{E}_{q(\boldsymbol{w}_{1:J})} \log \frac{p(\boldsymbol{y}_{1:J}, \boldsymbol{w}_{1:J}|\boldsymbol{\omega}, \boldsymbol{\sigma})q(\boldsymbol{w}_{1:J})}{q(\boldsymbol{w}_{1:J})p(\boldsymbol{w}_{1:J}|\boldsymbol{y}_{1:J}, \boldsymbol{\omega}, \boldsymbol{\sigma})} \tag{10}$$

$$= \underbrace{\mathbb{E}_{q(\boldsymbol{w}_{1:J})} \log \frac{p(\boldsymbol{y}_{1:J}, \boldsymbol{w}_{1:J}|\boldsymbol{\omega}, \boldsymbol{\sigma})}{q(\boldsymbol{w}_{1:J})}}_{ELBO(q(\boldsymbol{w}_{1:J}), \boldsymbol{\omega}, \boldsymbol{\sigma})} + D_{KL}\left(q(\boldsymbol{w}_{1:J})\|p(\boldsymbol{w}_{1:J}|\boldsymbol{y}_{1:J}, \boldsymbol{\omega}, \boldsymbol{\sigma}))\right).$$

$$\tag{11}$$

Since directly maximizing the marginal likelihood is computationally infeasible, we instead maximize its variational lower bound, commonly known as the evidence lower bound (ELBO):

$$ELBO(q(\boldsymbol{w}_{1:J}), \boldsymbol{\omega}, \boldsymbol{\sigma}) = \mathbb{E}_{q(\boldsymbol{w}_{1:J})} \log \frac{p(\boldsymbol{y}_{1:J}, \boldsymbol{w}_{1:J}|\boldsymbol{\omega}, \boldsymbol{\sigma})}{q(\boldsymbol{w}_{1:J})} \tag{12}$$

$$= \mathbb{E}_{q(\boldsymbol{w}_{1:J})} \log p(\boldsymbol{y}_{1:J}|\boldsymbol{w}_{1:J}, \boldsymbol{\omega}, \boldsymbol{\sigma}) - D_{KL}(q(\boldsymbol{w}_{1:J})\|p(\boldsymbol{w}_{1:J}|\boldsymbol{\omega}, \boldsymbol{\sigma})). \tag{13}$$

The measurements are mutually independent and do not depend on the latent variables $\{\boldsymbol{\omega}, \boldsymbol{\sigma}\}$ given network weights $\boldsymbol{w}_{1:J}$. Moreover, each network is trained solely on its corresponding measurement. Therefore, the log likelihood decomposes into: $\mathbb{E}_{q(\boldsymbol{w}_{1:J})} \log p(\boldsymbol{y}_{1:J}|\boldsymbol{w}_{1:J}, \boldsymbol{\omega}, \boldsymbol{\sigma}) = \sum_{j=1}^{J} \mathbb{E}_{q(\boldsymbol{w}_j)} \log p(\boldsymbol{y}_j|\boldsymbol{w}_j)$. Additionally, due to the factorized variational approximation $q(\boldsymbol{w}_{1:J}) = \prod_{j=1}^{J} q(\boldsymbol{w}_j)$ and the assumption of conditional independence $p(\boldsymbol{w}_{1:J}|\boldsymbol{\omega}, \boldsymbol{\sigma}) =$

$\prod_{j=1}^{J} p(\boldsymbol{w}_j | \boldsymbol{\omega}, \boldsymbol{\sigma})$, we can obtain the following form of ELBO:

$$ELBO(q(\boldsymbol{w}_{1:J}), \boldsymbol{\omega}, \boldsymbol{\sigma}) = \sum_{j=1}^{J} \mathbb{E}_{q(\boldsymbol{w}_j)} \log p(\boldsymbol{y}_j | \boldsymbol{w}_j) - D_{KL}(q(\boldsymbol{w}_j) \| p(\boldsymbol{w}_j | \boldsymbol{\omega}, \boldsymbol{\sigma})). \quad (14)$$

Since each network is trained separately, our framework runs efficiently via parallel computing. The likelihood $p(\boldsymbol{y}_j | \boldsymbol{w}_j)$ can be maximized by minimizing the squared error loss of the reconstruction (see Equation (1)). Similarly to Zhu et al. (2023), we adopt EM to maximize the ELBO. At the E-step, the latent variables $\{\boldsymbol{\omega}, \boldsymbol{\sigma}\}$ are fixed and each network optimizes with respect to the loss function:

$$\mathcal{L}(q(\boldsymbol{w}_j)) = \mathbb{E}_{q(\boldsymbol{w}_j)} \| \boldsymbol{A} \mathcal{F}_{\boldsymbol{w}_j}(\boldsymbol{C}) - \boldsymbol{y}_j \|_2^2 + D_{KL}(q(\boldsymbol{w}_j) \| p(\boldsymbol{w}_j | \boldsymbol{\omega}, \boldsymbol{\sigma})). \quad (15)$$

The expectation can be estimated via MC sampling. When the variational approximations $q(\boldsymbol{w}_j) \sim \mathcal{N}(\boldsymbol{\mu}_j, \boldsymbol{\rho}_j)$, $j = 1 \ldots J$ are formed, we can perform the M-step with the $q(\boldsymbol{w}_{1:J})$ being fixed. Simplifying Equation (14):

$$ELBO(\boldsymbol{\omega}, \boldsymbol{\sigma}) = -\sum_{j=1}^{J} D_{KL}\left( q(\boldsymbol{w}_j) \| p(\boldsymbol{w}_j | \boldsymbol{\omega}, \boldsymbol{\sigma}) \right) \quad (16)$$

$$\propto \sum_{j=1}^{J} \mathbb{E}_{q(\boldsymbol{w}_j)} \log p(\boldsymbol{w}_j | \boldsymbol{\omega}, \boldsymbol{\sigma}) \quad (17)$$

$$\propto -\sum_{j=1}^{J} \log | \operatorname{diag}(\boldsymbol{\sigma}) | + (\boldsymbol{\omega} - \boldsymbol{\mu}_j)^2 \cdot \boldsymbol{\sigma}^{-1} + \boldsymbol{\rho}_j \cdot \boldsymbol{\sigma}^{-1}, \quad (18)$$

where $\operatorname{diag}(\boldsymbol{\sigma})$ represent a diagonal matrix with the diagonal $\boldsymbol{\sigma}$. The above equation delivers a closed-form solution by setting the derivative to zero:

$$\frac{\partial}{\partial \boldsymbol{\omega}} ELBO(\boldsymbol{\omega}, \boldsymbol{\sigma}) \propto \sum_{j=1}^{J} \boldsymbol{\omega} - \boldsymbol{\mu}_j := 0 \quad \Rightarrow \quad \boldsymbol{\omega}^* = \frac{1}{J} \sum_{j=1}^{J} \boldsymbol{\mu}_j. \quad (19)$$

$$\frac{\partial}{\partial \boldsymbol{\sigma}} ELBO(\boldsymbol{\omega}, \boldsymbol{\sigma}) \propto \sum_{j=1}^{J} \boldsymbol{\sigma} - (\boldsymbol{\omega} - \boldsymbol{\mu}_j)^2 - \boldsymbol{\rho}_j := 0 \quad \Rightarrow \quad \boldsymbol{\sigma}^* = \frac{1}{J} \sum_{j=1}^{J} (\boldsymbol{\omega}^* - \boldsymbol{\mu}_j)^2 + \boldsymbol{\rho}_j. \quad (20)$$

## C  DETAILS FOR EXPERIMENTS

### C.1  INRWILD

We implement INRWild using the Siren network architecture (Sitzmann et al., 2020). Specifically, the static component, $\mathcal{G}_{\boldsymbol{\phi}}$, is represented using a standard 8-layer Siren network. In contrast, each of the transient parts, $\mathcal{H}_{\boldsymbol{w}_j}$, is characterized by a more compact 4-layer Siren network following the design from NeRFWild (Martin-Brualla et al., 2021). The optimization process ensures that the static and transient components are jointly optimized to ensure distinct representations, as articulated in Equation 2. A visual representation of the INRWild structure is provided in Figure 8.

### C.2  INR-BASED NETWORK CONFIGURATION

All INR-based methodologies in this study utilize the standard Siren (Sitzmann et al., 2020) as their foundational network. This network comprises eight fully connected layers, each with a width of 256. To encode position, we employed the Fourier feature embedding (Tancik et al., 2020). Every INR-based method features an embedding dimension of 512. Consistently, this embedded position is input to the INRs to facilitate density prediction.

### C.3  CONFIGURATION OF ALL METHODS

**Individual Reconstruction Methods.** For 2D experiments, we employ the FBP method, and for 3D experiments, its counterpart, FDK. GPU-accelerated operations, FBP_CUDA for 2D and FDK_CUDA

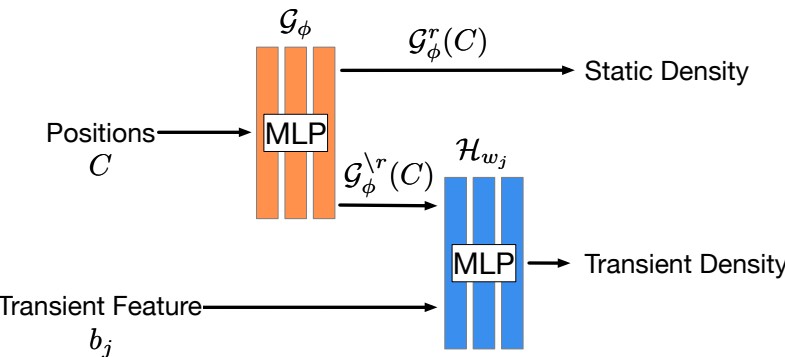

Figure 8: Schematic depiction of INRWild: A tailored version adapted from NeRFWild (Martin-Brualla et al., 2021) for joint CT reconstruction.

for 3D, are sourced from the Astra-Toolbox (Van Aarle et al., 2016). The iterative method `SIRT`, specifically the `SIRT_CUDA` operation, also from the Astra-Toolbox, is configured with 5,000 iterations. INR-based methods are set to operate for 30,000 iterations. The iteration counts for `SIRT` and INR-based methods are determined through preliminary tests on a dataset subset to ensure optimal performance.

**Joint Reconstruction Methods.** We tune the hyperparameters of our method and baselines in the inter-patient scenario and subsequently apply to all experiments. In particular, for the federated averaging approach `FedAvg` (Kundu et al., 2022), we average individual reconstructions every 300 iterations, amounting to a total of 100 averaging iterations. For the meta-learning technique `MAML` (Tancik et al., 2021), which is rooted in the MAML update policy (Nichol et al., 2018), we designate a learning phase spanning 10,000 iterations across all measurements. This is succeeded by 20,000 individual reconstruction iterations utilizing the learned parameters. Our method has the same iteration configurations as `FedAvg` with 100 global iterations to update the latent variables, each global iteration undergoes local 300 iterations across all nodes.

**Hyperparameters.** All INR-based models utilize the Adam optimizer Kingma & Ba (2014). The initial learning rate is set at $1 \times 10^{-5}$ for each node, with $\beta_1$ and $\beta_2$ values of 0.9 and 0.999, respectively. For the meta-learning phase in `MAML`, the inner learning rate is $1 \times 10^{-3}$, the outer learning rate is $1 \times 10^{-6}$, the inner steps are set at 10, and the outer steps at 1000. These settings were optimized based on a grid search conducted on a subset of the inter-patient scenario and were subsequently used across all experiments. For our method, the additional hyperparameter $\beta$ is determined as $1 \times 10^{-16}$. This was decided upon after evaluating the initial loss terms of Equation 4 from the same subset of the inter-patient scenario. This hyperparameter setting for our method is also applied consistently across experiments.

## D  DATASET DETAILS

**4DCT** This dataset Castillo et al. (2009), sized $10 \times 136 \times 512 \times 512$, contains 136 CT image slices captured across 10 respiratory phases of one patient. The main variations across these phases are due to respiratory movements, such as the lungs' expansion and contraction.

**LungCT** Comprising CT scans from 96 patients Antonelli et al. (2022), its volumes range from $112 \times 512 \times 512$ to $636 \times 512 \times 512$. Despite inherent similarities representing human lungs, individual scan features can vary significantly. Slices within a volume show a consistent pattern, yet fewer stationary features are shared between them. The dataset comprises scans both with and without tumors. For our experiments, we randomly selected patients and images without distinguishing between those containing tumors and those without, aiming for a diverse representation of lung CT images.

**BrainCT** The dataset Flanders et al. (2020), with 874,035 CT images of size $512 \times 512$, is annotated with hemorrhage labels. It's organized as image sets without specific patient linkages.

**CelebA** This dataset Li et al. (2021) consists of 202,599 celebrity face images of dimension $3 \times 218 \times 178$.

# E  EXTRA RESULTS

## E.1  3D JOINT RECONSTRUCTION

To assess the real-world viability of our method, we conduct evaluations in a 3D cone-beam CT context, which more closely aligns with practical scenarios. We choose CT volumes from different patients of size $128^3$, ensuring they represent analogous regions of the human body. The projections are simulated with 40 angles spanning a full $360°$ rotation.

We conducted experiments on 9 groups of joint reconstructions, with each group jointly reconstructing 10 different patients' CT volumes, each sized $128^3$. Table 3 displays the results. Consistent with the findings from 2D CT experiments, our approach surpassed other comparative methods, substantiating its practical relevance. `MAML` displayed slightly inferior performance compared to `SingleINR`. A potential rationale for this could be that, given the augmented data volume, meta-learning might necessitate extended meta-learning iterations to glean a meaningful representation.

|      | FDK | SIRT | SingleINR | FedAvg | MAML | INR-Bayes |
|------|-----|------|-----------|--------|------|-----------|
| PSNR | 19.40 $\pm0.62$ | 24.91 $\pm0.45$ | 33.99 $\pm0.42$ | 30.30 $\pm0.27$ | 33.67 $\pm0.55$ | **34.19** $\pm0.38$ |
| SSIM | 0.550 $\pm0.005$ | 0.650 $\pm0.007$ | 0.932 $\pm0.007$ | 0.862 $\pm0.004$ | 0.932 $\pm0.009$ | **0.945** $\pm0.004$ |

Table 3: Results from 3D cone-beam CT reconstruction. The highest average PSNR/SSIM values that are statistically significant are highlighted in bold.

## E.2  JOINT RECONSTRUCTION ON HUMAN FACES

To assess the versatility of our proposed method, we expand its application to an unconventional domain by considering natural RGB images as 3-slice objects, akin to CT. Although a departure from traditional CT contexts, this experiment aimes to test the methods' generalizability. For this purpose, we sample 10 distinct faces from the CelebA dataset (Liu et al., 2015) and project them using a parallel beam from 40 different angles.

The aggregate metrics for joint reconstruction across 10 groups, in total of 100 faces, are shown in Table 4. Notably, our approach surpasses other techniques by a considerable margin. This suggests not only its adaptability across diverse tasks but also underscores its potential applicability in other inverse problems, particularly where undersampling challenges reconstruction. A visual comparison in Figure 9 underscores our method's prowess, revealing richer details and minimizing reconstruction artifacts.

|      | FBP | SIRT | SingleINR | INRWild | FedAvg | MAML | INR-Bayes |
|------|-----|------|-----------|---------|--------|------|-----------|
| PSNR | 18.12 $\pm0.39$ | 29.13 $\pm0.20$ | 30.90 $\pm0.32$ | 16.43 $\pm0.28$ | 25.71 $\pm0.27$ | 30.74 $\pm0.29$ | **31.31** $\pm0.31$ |
| SSIM | 0.549 $\pm0.006$ | 0.766 $\pm0.005$ | 0.831 $\pm0.008$ | 0.260 $\pm0.011$ | 0.647 $\pm0.012$ | 0.827 $\pm0.007$ | **0.847** $\pm0.007$ |

Table 4: Results of joint reconstruction on human faces. The highest average PSNR/SSIM values that are statistically significant are highlighted in bold.

To offer a deeper insight, we visualize the learned priors across different joint reconstruction techniques. Figure 10 illustrates INRWild's extraction of "static" and "transient" components from varying faces. Notably, INRWild captured a generalized "face"-like static component. However, due to the significant disparities among face images, this generalized extraction do not significantly enhance individual reconstructions. Figure 11 showcases the learned priors from `FedAvg`, `MAML`, and our approach `INR-Bayes`. While `MAML` struggles to capture a face-like prior during its preliminary phase, both `FedAvg` and `INR-Bayes` succeed in deriving an interpretable prior. However, the averaging distinct faces does not directly benefit reconstruction in the case of `FedAvg`. In contrast,

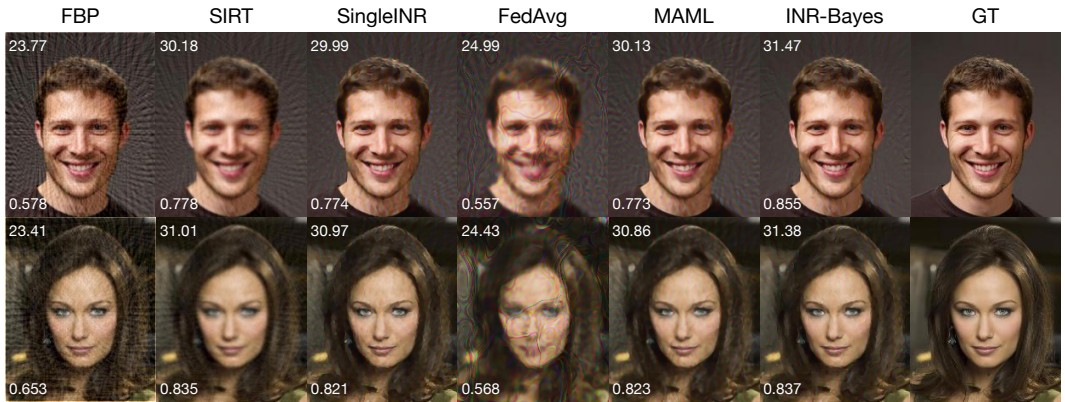

Figure 9: Visual comparison on human faces results. PSNR values are on the top left, with SSIM values on the bottom left.

our method astutely harnesses the cross-measurement prior information to amplify its individual performance.

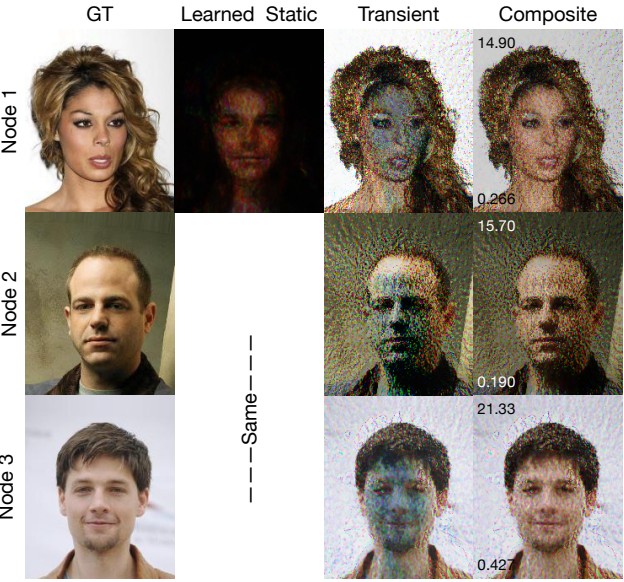

Figure 10: Learned static and transient parts of INRWild on human faces of CelebA dataset.

### E.3 Inter-Patient joint reconstruction

In this section, we provide a visual comparison of inter-patient joint reconstruction using the BrainCT dataset. As depicted in Figure 12, our approach consistently exhibits the best SSIM values, outperforming others by a significant margin. Although the PSNR values of our method are either the best or close to the best, it's worth noting that the ground-truth image, which we utilize to simulate projections, inherently contains a considerable amount of noise. This noise could potentially influence the performance metrics.

### E.4 Intra-patient joint reconstruction

In this section, we present visualizations of learned priors from various joint reconstruction methods applied in intra-patient experiments. Figure 13 depicts the learned static and transient components of INRWild. Notably, in scenarios where images markedly vary from one another, extracting static

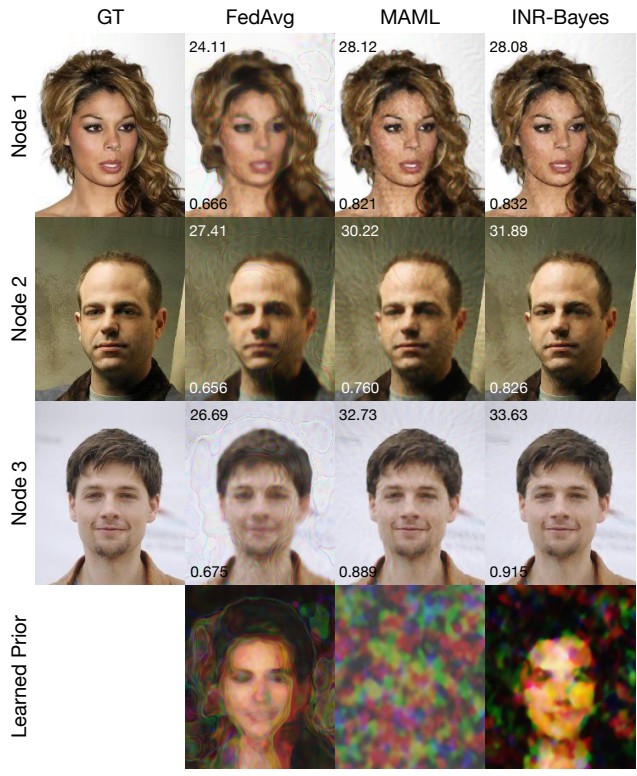

Figure 11: Visualization of learned prior of joint reconstruction methods on the faces of the CelebA dataset.

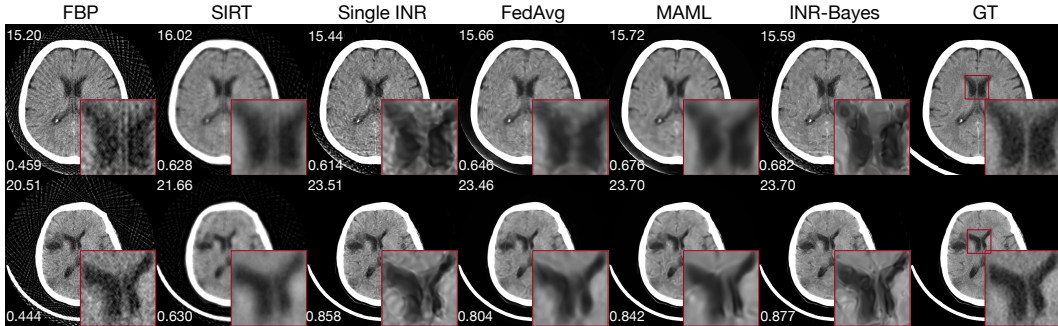

Figure 12: Visual comparison for joint reconstruction on brains of BrainCT dataset. Enlarged areas are highlighted in red insets. PSNR values are on the top left, with SSIM values on the bottom left.

components still seems feasible but not necessarily beneficial to the reconstruction process, since the static component does not constitute a significant portion of the overall representation.

Conversely, when observing joint reconstruction methods in Figure 14, it is evident that all joint reconstruction methodologies ascertain a reasonable meta representation. Despite variations in images, the intrinsic consistency stemming from the same patient results in a discernible and coherent trend. This inherent trend is adeptly captured by the joint reconstruction methodologies.

### E.5 JOINT RECONSTRUCTION ACROSS TEMPORAL PHASES IN 4DCT

Figure 15 showcases that INRWild is adept at differentiating between static and transient components. Nevertheless, INRWild's efficiency predominantly arises in scenarios where image variations are subtle. Such constraints limit its broader applicability in joint CT reconstruction. In contrast,

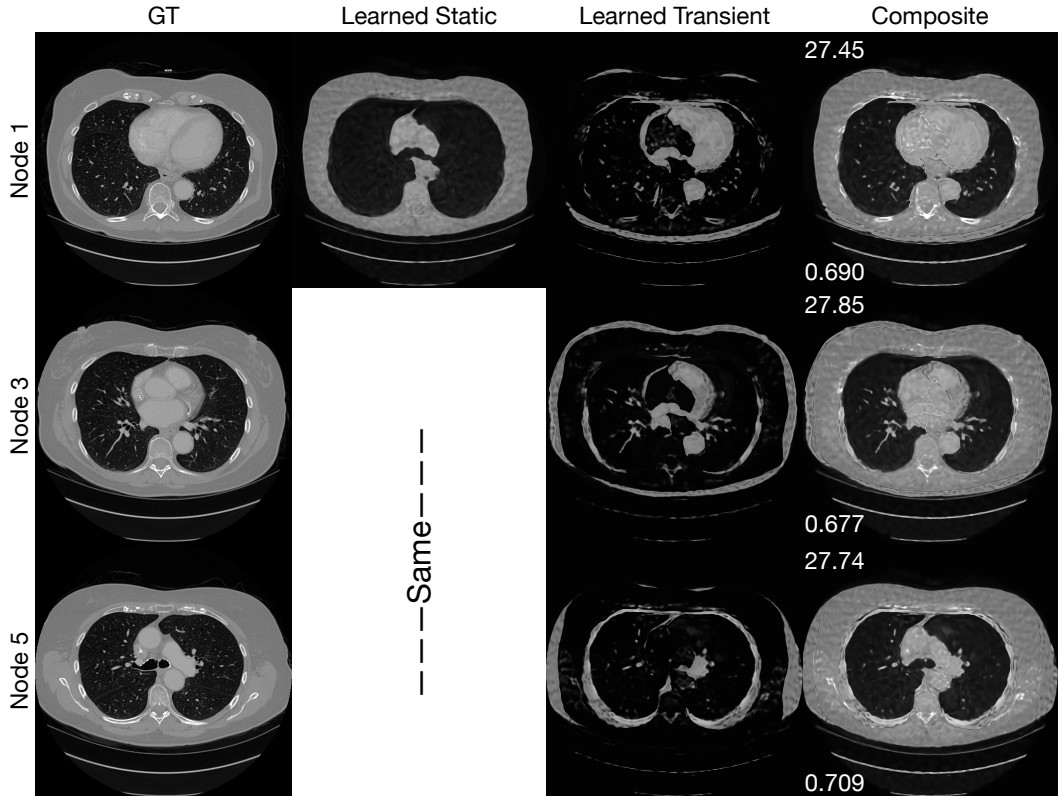

Figure 13: Learned static and transient parts of INRWild on LungCT dataset on the same patient.

Figure 16 demonstrates that other joint reconstruction methods also proficiently disentangle the inherent prior. Interestingly, in these contexts, an averaging approach proves more beneficial than meta-learned initialization. Notably, our proposed method continues to surpass both FedAvg and meta-learning in this particular scenario.

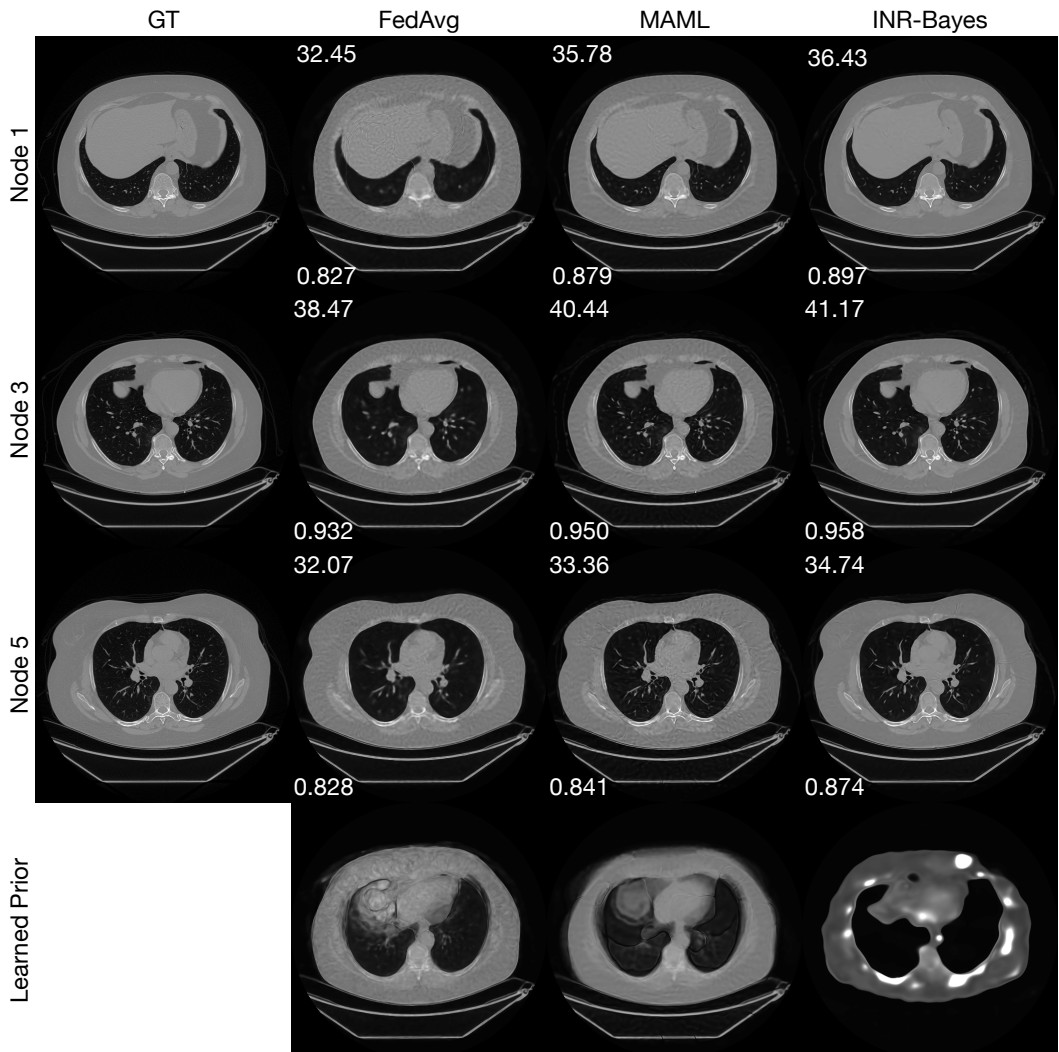

Figure 14: Visualization of learned prior of joint reconstruction methods on the same patient of LungCT dataset.

### E.6 COMPUTATION COST ANALYSIS

In Table 5, we present a comparative analysis of the computational costs associated with different reconstruction methods. The experiment setting is aligned the with inter-patient configuration in Table 1. These assessments were performed under identical conditions on the same workstation, equipped with an Intel I7-11700KF CPU and a single Nvidia RTX 3070 GPU, to ensure consistency in our evaluation.

|  | SingleINR | FedAvg | MAML | INR-Bayes |
|---|---|---|---|---|
| GPU Memory (MiB) | 6338 | 6408 | 6344 | 6452 |
| Time (hrs:mins) | 09:03 | 09:07 | 09:07 | 09:53 |

Table 5: Comparison of computation cost on joint reconstruction of 10 nodes. The reconstructed image size is $512 \times 512$. For reference, FBP requires 206 MiB of GPU memory and completes in 0.37 seconds, whereas SIRT utilizes 154 MiB of GPU memory and takes 13.62 seconds.

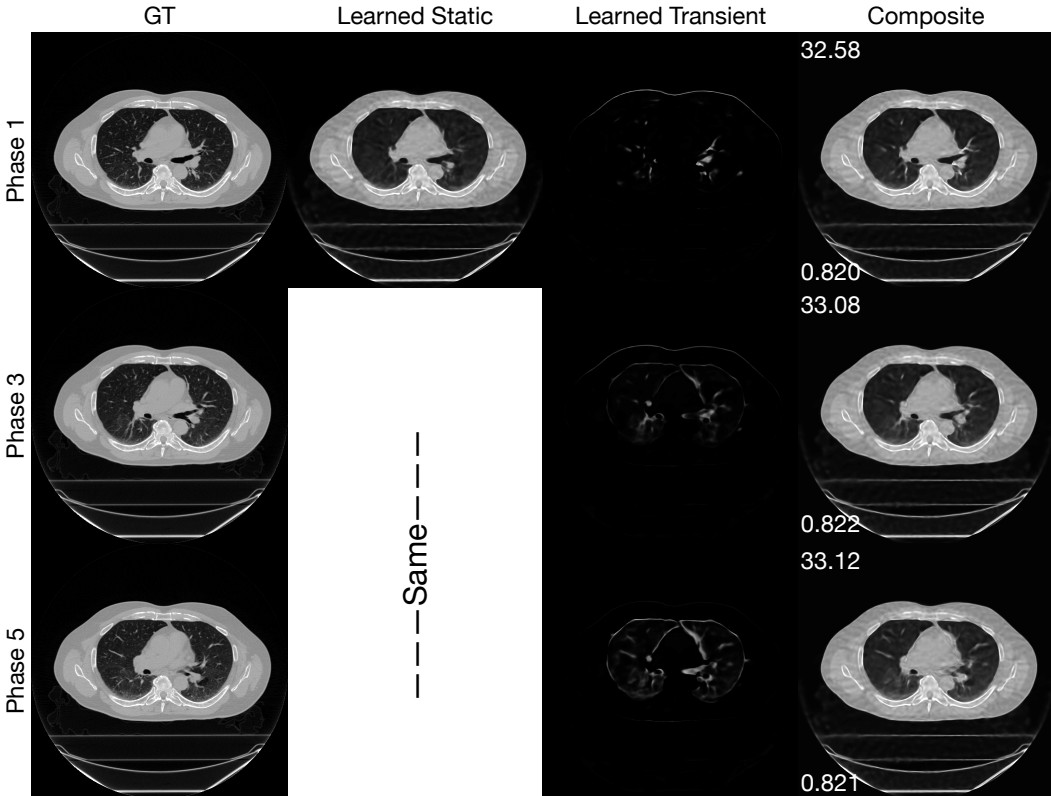

Figure 15: Learned static and transient parts of INRWild on 4DCT dataset.

As indicated in the Table 5, the GPU memory usage across all methods was relatively similar. `SingleINR` exhibited the shortest computation time, followed closely by `FedAvg` and `MAML`. Our method, `INR-Bayes`, showed a slightly increased computation time, approximately 50 minutes longer than the others. This less than $10\%$ increase over `SingleINR, FedAvg and MAML` in time is attributed to the added model capacibility and the Gaussian noise sampling procedure in `INR-Bayes`. However, considering the enhanced reconstruction quality and robustness achieved, this additional time investment can be justified.

### E.7 COMPARISON WITH NERP

The method `Nerp`, introduced by Shen et al. (2022), initially trains an INR network using high-fidelity data through regression. This pre-trained network is subsequently utilized to initialize the reconstruction of new object with sparse measurements. A notable drawback of this method is its dependence on the new objects' representations being highly similar to that of the high-fidelity training object. When this similarity is absent, the initial training could hinder rather than help the reconstruction process, potentially yielding worse results than even a random initialization.

To demonstrate this, we carried out experiments on the 4DCT dataset with two different setups for `Nerp`. In the 'Match' configuration, `Nerp` is provided with the ground truth of one phase at a specific slice and tasked to reconstruct the remaining nine phases at that slice. In contrast, the 'Unmatch' configuration uses the ground truth from a random slice. Our `INR-Bayes` approach, on the other hand, performs a simultaneous reconstruction of all nine phases without any access to ground truth images.

As Table 6 illustrates, the performance of `Nerp` is conditional, excelling in PSNR when ground truth data is matched but faltering otherwise. While operating without access to additional information, our `INR-Bayes` achieves the best performance in SSIM. Given the practical challenges in

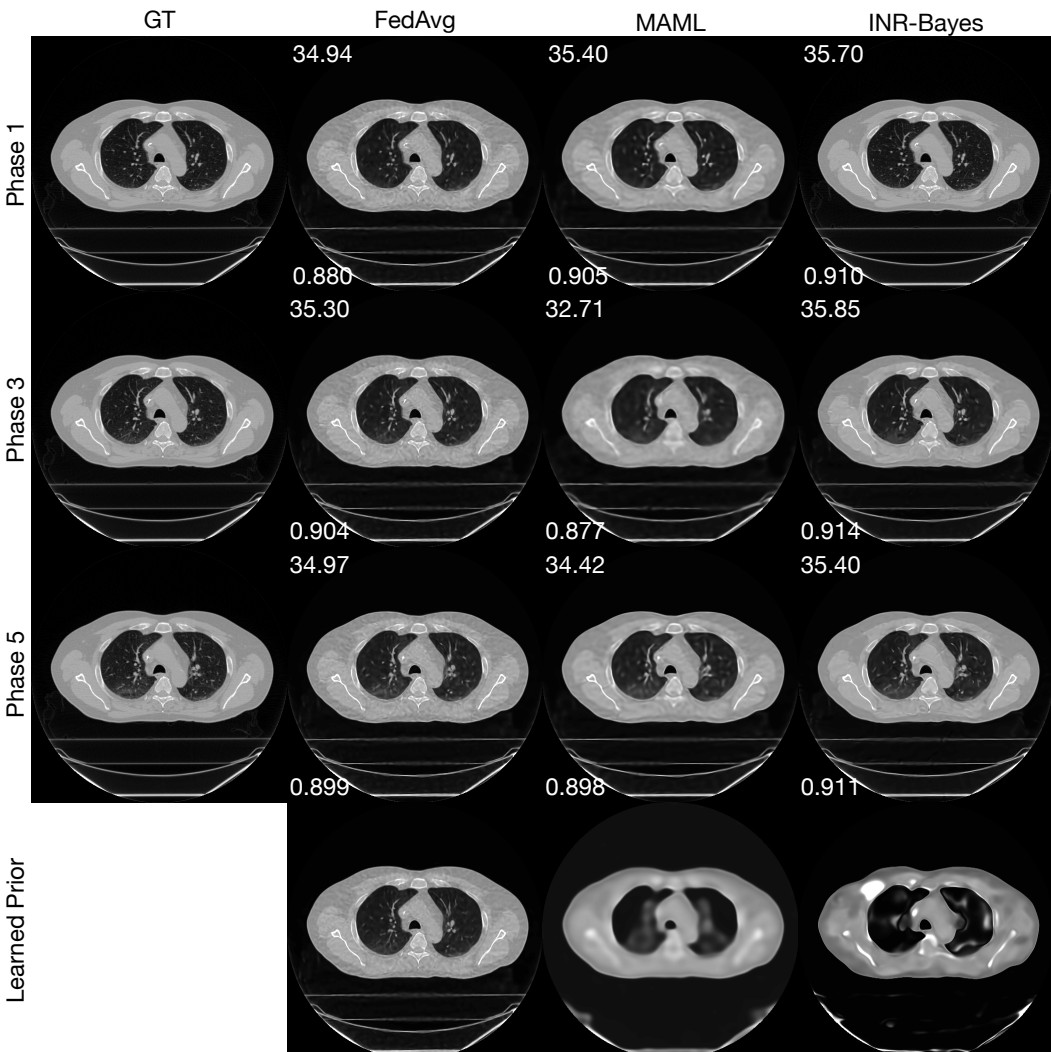

Figure 16: Visualization of learned prior of joint reconstruction methods on 4DCT dataset.

obtaining matched ground-truth data for unscanned objects, our method exhibits greater utility and applicability.

|  | SingleINR | Nerp match | Nerp unmatch | INR-Bayes |
|---|---|---|---|---|
| PSNR | 33.69 ±0.06 | **35.33** ±0.10 | 32.83 ±0.07 | 34.31 ±0.07 |
| SSIM | 0.883 ±0.002 | 0.889 ±0.001 | 0.849 ±0.02 | **0.901** ±0.001 |

Table 6: Results from 4DCT reconstruction. The highest average PSNR/SSIM values that are statistically significant are highlighted in bold.

## E.8 APPLYING TO UNSEEN DATA USING LEARNED PRIOR

**Impact of Different Priors on an Unseen Patient.** To investigate the influence of varying priors on the reconstruction quality for new, unseen patients, we conducted an additional experiment. We selected 10 sets of priors, each derived from a group of 10 different patients. These priors are then employed to guide the reconstruction of the same unseen patient. Figure 18 showcases the reconstructed images and their corresponding priors, represented by an INR that is parameterized with the mean of the prior distribution. The accompanying PSNR and SSIM values, indicated at

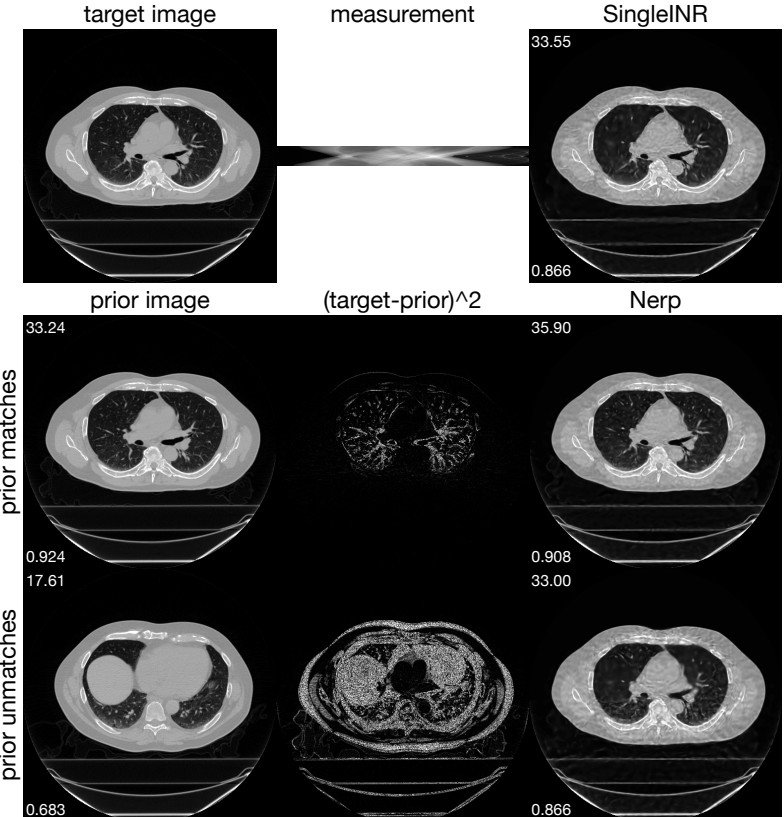

Figure 17: Results of Nerp with matched prior image and unmatched prior image. PSNR on the upper left corner and SSIM on the lower left corner are calculated with respect to the target image.

the top left and bottom left of each image, demonstrate modest deviation across different priors. Notably, no model collapse occurs despite the obvious visual difference in the prior means. This observation suggests that our method is stable and can effectively extract useful information from various priors when applied to unseen data.

It is important to clarify that the prior in Figure 18 is depicted using the mean of the prior distribution to parameterize an INR. However, this representation is an incomplete portrayal of the prior distribution's full characteristics. The variance associated with our method's estimates may contribute to the robust and effective utilization of the prior distribution, even when there are variances in the mean. This aspect of our model underscores its capability to leverage the entire prior distribution for stable performance.

**Impact of Joint Nodes Numbers on Learned Prior.** We expand the investigation to assess the influence of the learned prior derived from a varying number of nodes. The experiments are conducted using the inter-patient configuration. As depicted in Figure 19, our `INR-Bayes` demonstrates a slight performance enhancement as the number of nodes increases. `FedAvg` exhibits a similar trend, albeit with consistently lower performance compared to the other methods. Notably, `MAML` experiences a performance dip when scaled across a larger node ensemble, which has also been observed in Figure 4b.

These results indicate that `INR-Bayes` is capable of developing a more robust prior with contributions from an increased number of nodes. Interestingly, this upward trend is not present in the intra-patient configuration (c.f. Figure 4b), possibly because additional data from the same individual does not introduce significantly new information. In the current experiment, the prior for a new patient is learned from data of randomly selected other patients, thus the estimation incorporates inherent biases and randomness. As the number of contributing nodes grows, the mean and variance of the prior distribution are expected to converge to the population statistics, enhancing its benefit

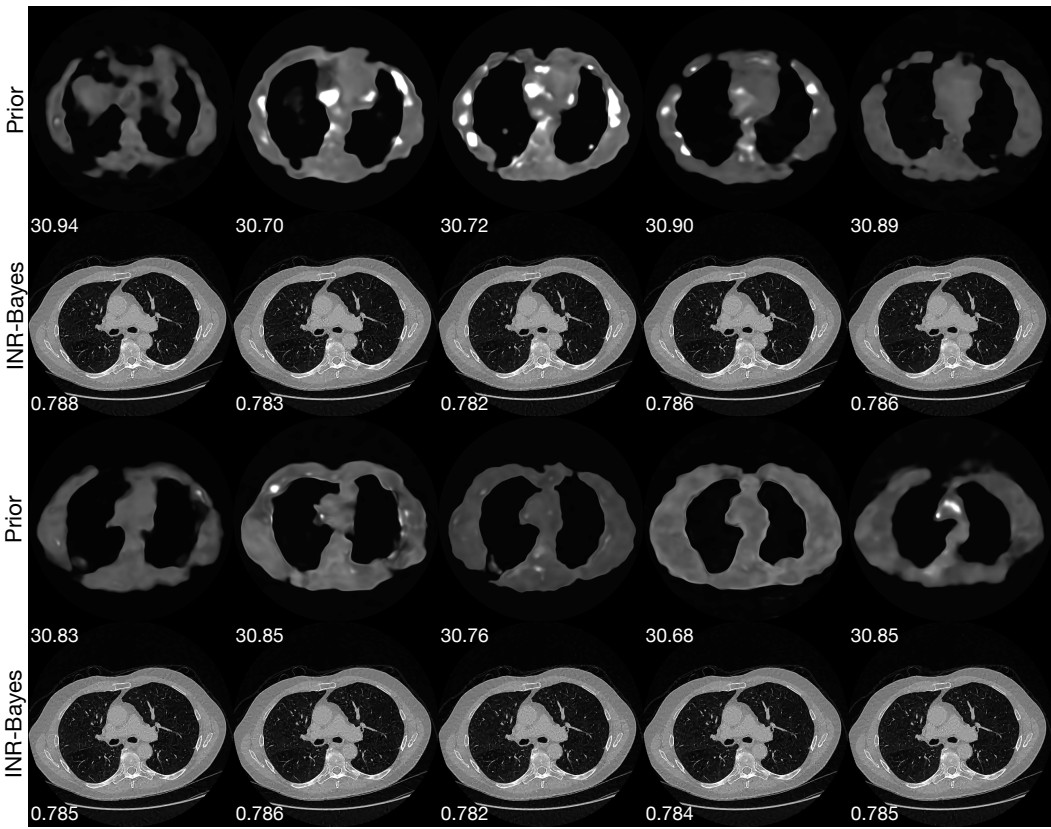

Figure 18: Reconstruction of the same unseen patient using different priors learned from various patient groups. The PSNR and SSIM values are presented on the top left and bottom left of each image, respectively, illustrating the method's robustness across different priors.

to the new patient. We anticipate that the advantage will continue to grow until it plateaus at a large number of joint nodes.

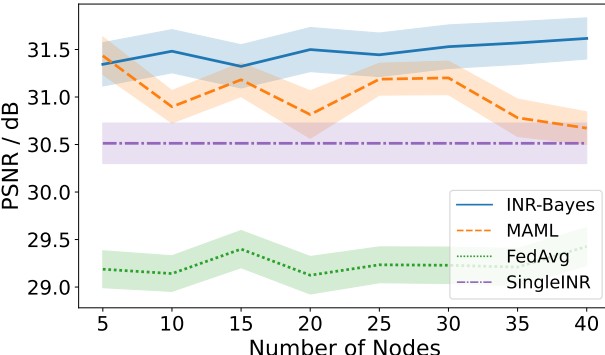

Figure 19: The performance of different methods on new patients using prior obtained with different number of patients. SingleINR is presented as reference.

