# OpenReview forum: "Implicit Neural Representations for Joint Sparse-View CT Reconstruction"
_ICLR.cc/2024/Conference — Submitted to ICLR 2024_

### Official Review · Reviewer_PdsV · 2023-10-29

**Soundness:** 3 good
**Presentation:** 3 good
**Contribution:** 3 good
**Rating:** 8
**Confidence:** 2

**Summary:**

This paper proposes a novel approach to improve reconstruction quality in Sparse-view Computed Tomography (CT) using Implicit Neural Representations (INRs) with a Bayesian perspective. The method incorporates latent variables to capture inter-object relationships and sets a new standard in CT reconstruction. The authors utilize three CT datasets and a natural image dataset to evaluate the generalizability of their approach. The proposed INR-based Bayesian framework enhances individual reconstructions and shows notably better metrics compared to other methods.

**Strengths:**

- Novel approach to improve reconstruction quality in sparse-view CT with INR, enhancing reconstructions.
- Extensive experiments and comparisons with other methods, to evaluate various facets of reconstruction performance
- Clear and detailed explanation of the methodology, including the EM algorithm and the alternating E and M steps used in the approach
- Results are well explained, each demonstrating the efficiency of including Bayesian framework.
- Well-designed and well-executed study that makes a significant contribution to the field of medical imaging.

**Weaknesses:**

No significant weakness in the paper.

**Questions:**

- For Figure 4b, there is a discussion that MAML might struggle to capture the shared features when many nodes are participating. Could you give bit detailed explanation? Also, which dataset are used for Figure 4? Was it 4DCT?
- Why would MAML fail to learn meaningful prior in Supplementary Figure 10?
- How does the proposed framework compare to other state-of-the-art methods in terms of computational efficiency?

nitpicks:
- Missing bold text in 5th row of Table 3.

---

> ### Author Response · Authors · 2023-11-17
> **Official Comments by Authors**
>
> Thank you for the thorough review and insightful feedback. We value the time you've dedicated to reviewing our work. In response, we have carefully prepared a rebuttal and revised the paper accordingly. Below, we address your questions:
>
> > Q1: For Figure 4b, there is a discussion that MAML might struggle to capture the shared features when many nodes are participating. Could you give bit detailed explanation? Also, which dataset are used for Figure 4? Was it 4DCT?
>
> In Figure 4b, we utilized an intra-person configuration on LungCT dataset from Decathlon. We have clarified this in the revision by clarifying it in the caption of Figure 4 “on intra-patient LungCT”. This additional detail should help readers better contextualize our findings.
>
> Furthermore, we conducted new experiments that applied priors learned from different numbers of nodes to unseen data using an inter-person configuration on LungCT dataset. The results (E8 and Figure 19) exhibit a consistent trend with MAML's performance.
>
> MAML is designed to identify an initial model that can be quickly fine-tuned to specific objects using a minimal number of gradient updates. Our hypothesis is that when the objects involved in joint training are excessively diverse, this initial model may not effectively generalize. Furthermore, while the original work [1] has successfully applied MAML to multiple objects in an image regression context, our empirical findings suggest that MAML struggles to develop a robust meta-model from many nodes when the reconstruction depends on projection measurements and the sampling is sparse.
>
> We appreciate the opportunity to delve deeper into these aspects of our research, as understanding the nuances of these methods is crucial for advancing the field. We believe these additional experiments and discussions will offer valuable insights to the readers and contribute to a more comprehensive understanding of the potential and limitations of various methodologies in the realm of CT reconstruction.
>
> > Q2: Why would MAML fail to learn meaningful prior in Supplementary Figure 10?
>
> In the visualization presented in Figure 10, it's observed that the learned prior of MAML does not depict a pattern that is immediately recognizable as a face, unlike those produced by FedAvg and INR-Bayes.
>
> It's important to clarify that this observation shouldn't necessarily be seen as an indication of MAML's inability to learn a meaningful prior. Firstly, the 'color lump' pattern associated with MAML is also discernible in the priors learned by our method, suggesting its potential relevance to the reconstruction process.
>
> Secondly, although FedAvg captures face-like prior, MAML's reconstruction results are quantitatively superior to those of FedAvg, as evidenced by the metrics presented alongside each image. This indicates that while the priors learned by MAML may not offer clear semantic interpretation visually, they contribute effectively to the task of reconstruction, which is the primary goal of these methods.
>
> We appreciate this opportunity to delve deeper into the nuances of how different methods approach and learn priors. Our goal is not just to compare these methods quantitatively but also to offer insights into their qualitative aspects.
>
>
> > Q3: How does the proposed framework compare to other state-of-the-art methods in terms of computational efficiency?
>
> Recognizing the relevance of this aspect, we have dedicated a new section in the appendix (E.6 and Table 5) to specifically address computation costs. This addition has also been made in response to similar queries from other reviewers. For your convenience, we have included Table 5 below, summarizing the computational costs of our method compared to other state-of-the-art approaches.
>
> |                    | SingleINR | FedAvg | MAML  | INR-Bayes |
> |--------------------|-----------|--------|-------|-----------|
> | GPU Memory (MiB)   | 6338      | 6408   | 6344  | 6452      |
> | Time (hrs:mins)    | 09:03     | 09:07  | 09:07 | 09:53     |
>
> As indicated in the table, the additional computational overhead of our INR-Bayes method, when compared to SingleINR and other methods, is relatively modest. The primary additional computational time in our method involves the calculation of the latent variable and one-time Monte Carlo sampling per iteration. Our approach is deliberately devised to keep the computational overhead minimal while still harnessing the benefits of our Bayesian framework.
>
> We hope this information adequately addresses your query regarding computational efficiency.
>
> > Q4:  Missing bold text in 5th row of Table 3.
>
> Thank you for highlighting this oversight. We have now corrected Table 1 in our manuscript to ensure consistency and accuracy. Your attention to detail is greatly appreciated.
>
> [1] Tanick et. al, Learned initializations for optimizing coordinate-based neural representations, CVPR 2021

---

### Official Review · Reviewer_R9JK · 2023-11-01

**Soundness:** 3 good
**Presentation:** 3 good
**Contribution:** 2 fair
**Rating:** 6
**Confidence:** 3

**Summary:**

This paper introduces a new method for improving the reconstruction quality of sparse-view CT scans using implicit neural representations (INRs). Addressing the challenges posed by undersampled data in Sparse-view CTs, this paper advocates for joint reconstruction of multiple objects/subjects, capitalizing on the shared information (statical regularities in the paper) often found in similar subjects. Central to their approach is an INR-based Bayesian framework that incorporates latent variables to discern inter-object relationships. These variables serve as a dynamic reference during the optimization process, ensuring enhanced reconstruction quality. This work achieves good results and promises to open source the code.

**Strengths:**

1. Novelty. I like the proposed approach to modeling shared information in a "nerf-in-the-wild" setting. It effectively extracts maximal shared information across different subjects and demonstrates its utility. This setting is new, at least to me. While some works have applied INRs to sparse-view CT challenges, this paper slightly sets it apart by jointly reconstructing multiple objects. Also, the incorporation of multiple latent variables in the Bayesian framework is a thoughtful addition, further enhancing the originality of the approach.

2. The paper is well-structured and offers an intuitive flow, making it easy to read and follow.

3. The tackled problem, sparse view CT reconstruction, holds its own significance in "AI+Med". The paper provides some extra (toy) examples in the appendix, which is appreciated.

**Weaknesses:**

1. Small improvement. While the proposed method is conceptually appealing, the performance improvement appears to be minimal. As illustrated in Table 1, the gains, though in the positive direction, are relatively slight. Such incremental progress might raise questions about the practical implications and advantages of adopting this new approach over existing methods.

2. Non-principled static-transient decomposition. From what I understand now, the current model seems to hinge on a static branch that remains uniform across all subjects, irrespective of their position in the 3D volume. While this might be appropriate in a NeRF-in-the-wild setting, given the fixed positioning of structures like buildings in the real world, its direct application to varied anatomical structures in the abdominal region seems problematic. Every patient's anatomy, although structurally similar, is unlikely to occupy the same 3D space due to innate variations (rigid/deformable transformations). Hence, applying this method without a template registration step appears misguided.

I think a more holistic solution might involve concurrently learning different deformation fields for individual subjects, and mapping each anatomical point to a unified canonical space. This would account for the inherent spatial variations between patients while ensuring a consistent reference frame for reconstructions.

**Questions:**

1. Increasing the number of nodes doesn't help much for almost all methods. Do the authors have any insights into this?

---

> ### Author Response · Authors · 2023-11-17
> **Official Comments by Authors**
>
> Thank you for the thorough review and insightful feedback. We value the time you've dedicated to reviewing our work. In response, we have carefully prepared a rebuttal and revised the paper accordingly. Below, we address your questions:
>
> > Q1: Small improvement. While the proposed method is conceptually appealing, the performance improvement appears to be minimal. As illustrated in Table 1, the gains, though in the positive direction, are relatively slight. Such incremental progress might raise questions about the practical implications and advantages of adopting this new approach over existing methods.
>
> We appreciate your perspective and would like to offer some additional context to highlight the significance of our results.
>
> Firstly, it's important to note that PSNR is a logarithmic metric, meaning that even seemingly small numerical improvements can reflect substantial enhancements in image quality. Specifically, our method shows an average improvement of around 1.5 dB over single INR and approximately 0.5 dB over MAML. In the context of PSNR measurements over 30 dB, these improvements are indeed noteworthy.
>
> Furthermore, in the “Applying to Unseen Data using Learned Prior” experiment, our method achieved an even more pronounced improvement – an increase of 0.9 dB over MAML and 1.4 dB over SingleINR. This further underscores the efficacy of our approach.
>
> Another crucial aspect of our method is its robustness against overfitting, a prevalent issue in INR-based approaches. Typically, these methods experience a decline in reconstruction quality after reaching a peak performance, making it challenging to set an appropriate early stopping criterion in practical scenarios. In contrast, our method not only consistently achieves superior reconstruction quality but also maintains this level once reached. This stability is a significant practical advantage over existing methods, offering more reliable and consistent results.
>
> We believe that these aspects collectively demonstrate the practical utility and advancements of our proposed method in the field of CT reconstruction.
>
> > Q2: Non-principled static-transient decomposition. From what I understand now, the current model seems to hinge on a static branch that remains uniform across all subjects, irrespective of their position in the 3D volume. While this might be appropriate in a NeRF-in-the-wild setting, given the fixed positioning of structures like buildings in the real world, its direct application to varied anatomical structures in the abdominal region seems problematic. Every patient's anatomy, although structurally similar, is unlikely to occupy the same 3D space due to innate variations (rigid/deformable transformations). Hence, applying this method without a template registration step appears misguided.
>
> Thank you for your insightful comments regarding the static-transient decomposition in our work. We acknowledge the importance of your observation, particularly in the context of CT reconstruction where anatomical structures can vary significantly in positioning and orientation.
>
> You are correct in noting that our implementation of INRWild, adapted directly from NeRFWild, treats the static branch uniformly across subjects. This approach, while effective in settings with fixed structural elements, may indeed present limitations when applied to anatomical structures that inherently exhibit spatial variations.
>
> While the datasets we used for our study do provide a basic level of standardization, with objects generally centered, there are certainly deviations. Interestingly, our method seems to mitigate these deviations to some extent, as the prior learning process inherently accounts for variations in the weight space when selecting patients randomly for joint reconstruction. Our experimental results suggest that this approach confers a degree of robustness against the deviations present in our datasets.
>
> However, your point about the potential benefits of incorporating a registration step is well taken. Indeed, integrating our method with a template registration step could further enhance the accuracy and applicability of our approach, especially in cases with significant anatomical variations. This is an interesting direction for future research, and we are excited to explore the potential improvements this could bring to our method.
>
> Thank you again for your constructive feedback and for highlighting this important aspect of our work. We are eager to delve into this area in our future research endeavors.

---

> ### Author Response · Authors · 2023-11-17
> **Official Comments by Authors**
>
> > Q3: Increasing the number of nodes doesn't help much for almost all methods. Do the authors have any insights into this?
>
> The experiment depicted in Figure 4b was conducted using an intra-patient setup. Our hypothesis is that the statistics of a single patient's data can be effectively captured with just a few slices, and that adding extra nodes does not significantly improve the estimation. This phenomenon is likely to be consistent across different methods that extract statistical information from the INR network.
>
> We have included a new experiment in the Appendix (E8 and Figure 19), conducted in an inter-patient setup. In this scenario, extracting statistics from a larger number of nodes should more accurately converge to the population statistics. Consequently, we observed a slight improvement in our method's performance as the number of nodes increased.

---

### Official Review · Reviewer_WCTi · 2023-11-01

**Soundness:** 2 fair
**Presentation:** 3 good
**Contribution:** 2 fair
**Rating:** 5
**Confidence:** 5

**Summary:**

This paper proposes a method to implicit neural representation learning (INR) for joint sparse-view CT reconstruction, which means that to reconstruct several CT images at the same time. The proposed method is evaluated on different CT image datasets and shows better performance compared with previous meta-learning based INR methods.

**Strengths:**

- The proposed method may investigate an interesting research question that how to incorporate population priors in INR learning. Although I do have quite a few concerns and questions about the proposed method as below, the proposed INR-Bayes method may be a potential way by introducing latent variables so that it may be possible to make it as a generative model from some prior distribution in some way..
- The paper validates the proposed method on different CT image datasets with different CT configuration or settings, and compares them with different baselines. The experiments about adaptation on new patients using priors learned from other patients are an interesting setting, but may also be questionable as follows.

**Weaknesses:**

- The motivation for conducting joint CT reconstruction. From the perspective of clinical applications, I do not see any reason why we want to do joint CT reconstruction. To my best knowledge, there are no such settings and needs from current clinical protocol. Can we imagine that in a scenario, after one patient is scanned, we do not do the reconstruction right after the scanning but wait until there are 5-10 patients’ scans, then we want to do the reconstruction together? I cannot think about some applications that require such needs, maybe the author can explain more or give some specific examples.
- What is the physical meaning of the learned priors? This prior is learned from 10 slices (which mimics 10 different patients). From this setting, I guess the prior may be some “average” CT image across these 10 images including mostly low-frequency signals. This guess is also supported by the illustration of learned prior in Figure 13 and 15, which is somewhat the general structure of the sliced anatomic structure. But why should this prior be helpful to reconstruct higher quality of CT for new patients? In the sparse-view CT reconstruction, due to the sparse sampling, what is always missing is the high-frequency signal in the detailed structure. Why does such an “average” image should be helpful to improve the final reconstruction image quality to get sharper and fine structures?
- How to choose these 10 patients to get the prior? If we consider a setting to use the learned prior for new patients’ reconstruction, how shall we choose the 10 different patients to get the prior? Such as healthy patients or abnormal patients? For example, if there are some patients with tumors, does the learned prior also include such prior in the latent variables and indicate that in the new patients’ reconstruction? Shall there be any relationship between the new patients and prior patients? How can we know if the new patient is normal or abnormal before we get the CT image reconstructed?
- How does the method deal with registration problems in CT imaging? The validated datasets in the paper seem to be already registered. If we consider the real CT scanning in practice, for different patients, the patient’s positions will always be different. How can this method deal with the position shift when learning the prior from different patients?
- In the motivation as well as experiments, one important baseline that the paper compares with is MAML [1]. As the author also mentioned, [1] learned an initialization from multiple objects in order to speed up optimization process while cannot achieve better optimization results from the learned initialization. The proposed works share a lot of similarity with [1] while using a different way to formulate and parametrize the learned prior, why the proposed method would achieve better optimization results while [1] not. The results in Table 1 also support these where the scores for these two methods are quite comparable. Besides, does the INR-Bayes and MAML use the same encoding function, embedding size and backbone network structure in this comparison?
- In the previous works, the paper mentioned “Lastly, while alternative joint CT reconstructions like (Shen et al., 2022) [2] use priors from pre-reconstructed images, and (Reed et al., 2021) [3] relies on finding a template image from 4DCT; their practical limitations led to their exclusion from our comparative analysis.” First, to my understanding, these two works are not doing the joint CT reconstruction as claimed in this paper. [2] is doing the CT and MRi reconstruction through INR by using a full-sampled prior image of the same patient as prior embedding, which is a very common setting for patients’ longitudinal study in clinics. [3] is doing dynamic CT reconstruction where different frames share some similarity while maintaining deformable motion, which is also very common in 4DCT setting with motion. Second, I do not see what is the “practical limitations led to their exclusion from our comparative analysis”, since these two papers’ setting may be more reasonable from practical applications. And their goal is to achieve better reconstruction results instead of fast convergence as [1], so I think these two papers may even be more important to be  compared with to demonstrate the superiority of the proposed method.
- In the setting of “Applying to Unseen Data using Learned Prior”, how are the patients chosen to learn the prior? Would different prior patients influence the reconstruction for the same new patient? Would different number of prior patients influence the reconstruction for the same new patient? When adapt the new patients, will the latent variable also adapt to the new patient?
Computational efficiency. Based on Algorithm 1, it seems that multiple networks are maintained and trained simultaneously for different patients. Also it needs to iterate through all patient, all time steps, with three loops interleaved. This algorithm looks very costly for memory and time efficiency. Can the paper report the memory and time used in training and testing with comparison of baseline methods?
- Using some framework figure may be better to illustrate the whole framework.

[1] Learned initializations for optimizing coordinate-based neural representations. In Proceedings of the IEEE/CVF Conference on Computer Vision and Pattern Recognition, pp. 2846–2855, 2021.

[2] Nerp: implicit neural representation learning with prior embedding for sparsely sampled image reconstruction. IEEE Transactions on Neural Networks and Learning Systems, 2022.

[3] Dynamic ct reconstruction from limited views with implicit neural representa- tions and parametric motion fields. In Proceedings of the IEEE/CVF International Conference on Computer Vision, pp. 2258–2268, 2021.

**Questions:**

Please see weakness for the details of questions.

---

> ### Author Response · Authors · 2023-11-17
>
> Thank you for the thorough review and insightful feedback. We value the time you've dedicated to reviewing our work. In response, we have carefully prepared a rebuttal and revised the paper accordingly. Below, we address your questions:
>
> > Q1:  Application examples of joint reconstruction?
>
> We appreciate the opportunity to clarify the practical motivations behind our approach to joint CT reconstruction. To address this, we have now included the sentence 'Computed Tomography (CT) plays a crucial role in both medical diagnostics and industrial quality control.' as the first sentence in our abstract.
>
> Our study aims to explore the potential benefits of joint reconstruction in scenarios where multiple CT scans share similarities, either in a medical or industrial context. We have added the sentence, 'Computed Tomography (CT) plays a crucial role in both medical diagnostics and industrial quality control,' to the abstract for clearer emphasis on its significance. In medical settings, patients can sometimes undergo multiple CT scans over extended periods, particularly for conditions like evolving tumors. Joint reconstruction can significantly enhance the consistency and quality of these serial scans. This approach could be particularly beneficial in situations where the same anatomical regions are scanned repeatedly, allowing for improved image quality and potentially lower radiation doses by using sparse-view scans.
>
> In addition, different hospitals or CT machines could potentially use scans for joint reconstruction. The concept of applying previously learned information to new, unseen data presents another practical use. In such cases, we can store and leverage past reconstruction data to enhance future scans.
>
> While our research primarily evaluates medical images, the principles of joint reconstruction are equally applicable in industrial CT settings. For example, in assembly lines where similar objects are scanned routinely, batch processing of scans can efficiently utilize joint reconstruction techniques. This approach can lead to improved quality and consistency in industrial quality control processes.
>
> We acknowledge that the direct clinical application of joint reconstruction as described might not be a current standard practice. However, our research opens avenues for future exploration and potential implementation in medical, industrial, and scientific fields. We believe our findings contribute significantly to the advancement of CT imaging techniques and their potential applications.

---

> ### Author Response · Authors · 2023-11-17
> **Official Comments by Authors**
>
> > Q2: Physical meaning of the learned prior? Why can such prior help to reconstruct higher quality of CT for new patients? In the sparse-view CT reconstruction, due to the sparse sampling, what is always missing is the high-frequency signal in the detailed structure.
>
> The learned prior in our Bayesian method is modeled as a Gaussian distribution in the weight space of the INR. We use its mean to illustrate that the prior captures significant anatomical information in Figure 13 and 15. However, we emphasize that this is only one aspect of its role. The full potential of the prior encompasses the representation of uncertainty, distinguishing between universally observed patterns and those unique to fewer or individual subjects. This uncertainty allows for adaptive regularization during individual training, as each network autonomously determines which information to extract from the prior based on its own uncertainty and that of the prior. This process is facilitated by minimizing the KL divergence between the prior and the posterior distribution of the individual network. Such adaptability distinguishes the Bayesian framework from other methods and underpins its superior performance.
>
> In sparse-view CT, the challenge predominantly arises from scanning at fewer angles compared to traditional CT. This often leads to the Nyquist-Shannon sampling theorem not being fully met, which can result in compromised reconstruction quality. Specifically, there's a perceived loss of high-frequency information in the reconstructed images. However, it's crucial to note that the high-frequency information isn't inherently lost during scanning; rather, it's the insufficiency of scans (or sampling) that impedes the accurate reconstruction of these high-frequency signals.
> To demonstrate that sparse sampling doesn’t cancel out high frequency information, and with help of knowledge about low-frequency information we can reconstruct the high-frequency information, we design the following toy example. Consider a signal in the form y = sin(px) + cos(qx), where the coefficients p and q satisfy p, q \in [0, 1] and p < q. This inequality indicates that the signal comprises both high-frequency and low-frequency components. With sparse sampling, we obtain limited data, for instance, we know that when x = \pi, y = -0.5. From this single measurement, we cannot independently reconstruct the values of p or q. However, if the value of p (representing the low-frequency component) is known, we can deduce the value of q (representing the high-frequency component) from the sparse sample. Assuming p = 1/6, the equation becomes:
> sin(\pi) +cos(q\pi) = -0.5.
> Since sin(\pi/6) = 0.5, this equation simplifies to:
> cos(q\pi) = -1,
> which leads to the solution q = 1.
>
> This example illustrates how knowledge about low-frequency components can guide the reconstruction of high-frequency details, even with sparse sampling. Our method leverages this principle, utilizing the prior to fill in the missing information and thus enhance the overall reconstruction quality.
>
> > Q3: How to choose 10 patients to get the prior? Such as healthy patients or abnormal patients? For example, if there are some patients with tumors, does the learned prior also include such prior in the latent variables and indicate that in the new patients’ reconstruction? Shall there be any relationship between the new patients and prior patients? How can we know if the new patient is normal or abnormal before we get the CT image reconstructed?
>
> In our experiments, we don’t make specific requirements on the prior patients. For 4DCT, it’s natural to use all 10 phases to jointly reconstruct as they have the common changing trend. For intra-patient, we randomly select slices along the longitude, they also share the common changing trend as they are sampled from the same patient. For inter-patient experiments on Medical Segmentation Decathlon and Brain CT Hemorrhage Challenge, we randomly choose those prior slices. As they are representing the same area, lung or brain, they shall share some similarities which can be captured by the prior of our method. Specifically, the lungCT dataset of  Medical Segmentation Decathlon contains both images with tumor and without tumor, but we don’t make use of that information and randomly select 10 patients from the dataset, so the learned prior shall work for both normal and abnormal cases. Results show that our method is robust to the variation of these factors, suggesting that our method can improve the reconstruction quality without making additional requirements in normal CT applications.
>
> We added sentences “The dataset comprises scans both with and without tumors. For our experiments, we randomly selected patients and images without distinguishing between those containing tumors and those without, aiming for a diverse representation of lung CT images.” in the Dataset details section (D) to better clarify this point.

---

> ### Author Response · Authors · 2023-11-17
>
> > Q4: How does the method deal with registration problems in CT imaging? … How can this method deal with the position shift when learning the prior from different patients?
>
> Our Bayesian framework and meta learning baselines like FedAvg and MAML learn the prior in the parameter space of an INR network. The INR network architecture itself may provide the ability to mitigate the registration problem. Although the datasets we used indeed provide images that are reasonably centered, there are always deviations in terms of positions of different patients and their sizes. In our inter-person experiments, we randomly select patients for joint reconstruction, therefore the deviation emerges in the learning process. However, the results show that our method and some baselines are robust to this potential issue. A joint optimization with a registration method and our method may further improve performance. However, the study of registration methods is orthogonal to our research on our Bayesian framework. Combining with registration methods may be an important step further towards the practical usage, which can be explored in future work.
>
> > Q5: … why the proposed method would achieve better optimization results while [1] not. The results in Table 1 also support these where the scores for these two methods are quite comparable.
>
> MAML learns a meta-model for rapid adaptation to various objects. In our Bayesian framework, distributions are introduced to the network weights and the prior, with the latter acting as a regularizer in the optimization objective. By incorporating manually set early stopping criteria, MAML can be cast as conducting a Maximum A Posteriori (MAP) reconstruction [4]. In this respect, it bears some resemblance to our Bayesian framework. However, key distinctions exist between these two methods. Firstly, our Bayesian framework explicitly sets a Gaussian distribution for the network weights and prior, while the prior of MAML is induced by the early stopping in an implicit manner. A Gaussian distribution is a natural and effective choice for the weight distribution [5]. Secondly, to cast MAML as a Bayesian framework, the integration of individual network weights is assumed to be approximated by making use of a point estimate [4]. Lastly, setting good early stopping criteria is difficult in practice without access to the ground truth image, whereas our Bayesian framework does not require this hyperparameter.
>
> Our experimental results demonstrate that our method outperforms MAML, especially when applying the learned prior to unseen data (see Table 2 in the paper). Additionally, while INR methods commonly face overfitting issues as Figure 5 shows, and setting early stopping criteria is challenging without access to a ground-truth image, our method shows robustness to overfitting. The performance consistently reaches and maintains its peak after sufficient training, highlighting a critical practical advantage of the Bayesian approach.
>
> > Q6: Besides, do the INR-Bayes and MAML use the same encoding function, embedding size and backbone network structure in this comparison?
>
> For all comparison methods in our paper, we used the same encoding function, embedding size, total number of optimization iterations, and backbone network structure, ensuring a fair comparison. We have revised Section 5 to make this aspect clearer.

---

> ### Author Response · Authors · 2023-11-17
>
> > Q7: Comparison with [2] and [3]
>
> The works [2][3] also leverage information from (an)other object(s) to assist in the reconstruction of a single object. Classifying them as joint reconstruction methods may lead to some confusion, so we have updated the revision by changing the statement to “Lastly, while other INR-based CT reconstruction methods exist…”
>
> Regarding [2], you are right in noting its practical approach where a fully-sampled reconstructed image of the same patient is used as a prior. This method is indeed intuitive and aligns well with common clinical practice for longitudinal studies. However, the challenge arises when the prior image does not closely match the target image. To illustrate this, we conducted an additional experiment, the results of which are presented in Figure 17. This experiment demonstrates that when there is a mismatch between the prior and target images, the performance of [2] can be inferior even to SingleINR with random initialization. The table below compares [2] with our method in different scenarios, including matched and unmatched priors. [2] achieves better PSNR with matched prior, but its SSIM is still lower than our method. In addition, all experiments in our paper only access a set of sparse measurements. Providing a fully-sampled prior image to Nerp can therefore be considered an unfair comparison.
>
> |                | SingleINR             | Nerp match             | Nerp unmatch          | INR-Bayes              |
> |----------------|-----------------------|------------------------|-----------------------|------------------------|
> | PSNR           | 33.69 ± 0.06          | **35.33**  ± 0.10      | 32.83 ± 0.07          | 34.31 ± 0.07           |
> | SSIM           | 0.883 ± 0.002         | 0.889 ± 0.001          | 0.849 ± 0.02          | **0.901** ± 0.001      |
>
>
> As for [3], its focus on dynamic CT and the use of deformable motion are indeed relevant to our 4DCT experiment. However, this specific approach may not be directly applicable to other experimental settings presented in our paper. Moreover, their codebase is prohibited from release by their sponsor [https://github.com/awreed/DynamicCTReconstruction]. Despite reaching out to the authors, we have not yet received any response that would enable us to conduct comparison experiments with their approach. We appreciate your comments which have helped us refine our paper and re-evaluate our comparison methods. We believe our additional experiments and the ensuing discussion better contextualize our method’s advantages and limitations.

---

> ### Author Response · Authors · 2023-11-17
> **Official Comments by Authors**
>
> > Q8: In the setting of “Applying to Unseen Data using Learned Prior”, how are the patients chosen to learn the prior? Would different prior patients influence the reconstruction for the same new patient? Would different number of prior patients influence the reconstruction for the same new patient? When adapt the new patients, will the latent variable also adapt to the new patient?
>
> Thank you for your insightful questions regarding the selection of patients for learning the prior and its influence on reconstructing new patients. We appreciate the opportunity to clarify these aspects and extend our analysis.
>
> For the experiment mentioned in the main paper, we selected patients randomly, ensuring a consistent slice position across different patients by proportionally selecting slices relative to the total volume. This method accommodates the variation in the number of slices and volume sizes across patients, which can range from (112,512,512) to (636,512,512). In this particular experiment, the prior is not updated with data from the new patient. However, we acknowledge the potential for further enhancing the prior by including the new patient's data.
>
> Addressing your query on the influence of different prior patients on the reconstruction of the same new patient, we extended our experiments (Appendix F.8). In this new experiment, we trained the prior using 10 different groups of patients, each group consisting of 10 randomly selected patients. We then used these 10 different priors to guide the reconstruction of the same new patient. As illustrated in Figure 19, our method effectively utilized these diverse priors to reconstruct the new patient, demonstrating robustness against variability in the selected prior patients.
>
> Regarding your question about the impact of the number of prior patients, we conducted another experiment to investigate this. We trained our prior using varying numbers of patients, ranging from 5 to 40, and then applied these priors to the reconstruction of the same 5 new patients. The results, presented in Figure 18, indicate a gradual improvement in PSNR when employing priors learned from a larger pool of patients. This finding suggests that having priors from more patients can be beneficial for our method, enhancing the reconstruction quality.
>
> We hope these additional experiments and their results comprehensively address your queries and provide further insights into the robustness and adaptability of our proposed method.
>
> > Q9: Computation cost.
>
> We understand that the algorithm's complexity and its implications on memory and time efficiency are crucial considerations.
>
> While our methods iterate over multiple patients, SingleINR also needs to run on each of the patients in order to obtain their reconstructions. Figure 5 shows that all methods converge at a similar speed. In all experiments, we run all methods for 30000 iterations and compare their performance. While our method does incur additional computational complexity due to the optimization of the latent variables, we derive an efficient algorithm to achieve that (Line 6, 7 in Algorithm 1). The increase of the computational time of our method is therefore less than 10% compared to SingleINR. However, considering the enhanced reconstruction quality and robustness to overfitting achieved, this additional time investment can be justified.
>
> To clarify the computational efficiency, we have expanded our appendix to include a detailed analysis of the computational cost. We've also included a comparative table (Table 5) in this discussion for easy reference. You can find this detailed comparison [here](https://openreview.net/forum?id=vyGp9Mty2t&noteId=CI64s0jmxF).
>
> We hope this additional information adequately addresses your query and provides clarity on the computational aspects of our method.
>
> > Q10: Using some framework figure may be better to illustrate the whole framework.
>
> Thank you for your feedback, we agree that a visual representation could enhance the clarity and understanding of our methodology.
>
> While we believe that Figure 1 in our paper already offers a direct insight into the technical aspects of our method, we understand the importance of a more comprehensive visual overview. To address this, we have added a more detailed version of the method overview as Figure 7 in the appendix.
>
> We hope that this additional figure will provide readers with a clearer understanding of our framework, especially before they delve into the more technical details of our approach. We are also open to further modifying the overview figure to make it even more comprehensive, should you have specific suggestions or comments in this regard.

---

> ### Author Response · Authors · 2023-11-17
> **Official Comments by Authors**
>
> We appreciate your insightful feedback and have diligently addressed the concerns raised. We hope that our revisions and clarifications demonstrate the robustness and relevance of our work, and we respectfully invite you to re-evaluate our submission in light of these updates.

---

> ### Author Response · Authors · 2023-11-17
> **Official Comments by Authors**
>
> [4] Grant et al., Recasting Gradient-Based Meta-Learning as Hierarchical Bayes, ICLR, 2018.
>
> [5] Matthews et al., Gaussian Process Behaviour in Wide Deep Neural Networks, ICLR, 2018.

---

> ### Author Response · Authors · 2023-11-21
> **Rebuttal follow up**
>
> As we are now approaching the end of the reviewer-author discussion
> period, we would like to follow up on our rebuttal response and gently
> remind you to provide your valuable feedback.
>
> We are keen to know if our response and improvements address your
> concerns and satisfy you. Furthermore, we would like to confirm if there
> are any further concerns or questions you may have regarding our work,
> we are more than willing to engage in a productive discussion during the
> reviewer-author period.
>
> Thank you for your valuable input and time.

---

> > ### Author Response · Authors · 2023-11-22
> > **Rebuttal follow up**
> >
> > Dear Reviewer WCTi:
> >
> > We want to express our gratitude for the time and effort you have invested in reviewing our work. Your insights have been invaluable in enhancing the quality of our work. As the author-reviewer discussion period is closing soon, we would like to kindly remind you of the rebuttal we submitted addressing the concerns raised in your initial reviews. We have made a concerted effort to thoroughly address each point and believe that our responses might positively impact your evaluation of our paper.
> >
> > Given the importance of your feedback in the decision-making process and the upcoming deadline, it would be greatly appreciated if you could review our responses at your earliest convenience. We remain at your disposal for any last-minute clarifications and look forward to your final evaluation. Thank you once again for your dedication and valuable insights.

---

> ### Comment · Reviewer_WCTi · 2023-11-23
>
> Thanks for author's efforts to answer my questions. I have read the author's response and other reviewer's comments. Some of my concerns are addressed while I am still not fully convinced with other questions including:
>
> Q1: It is still not clear in what kind of applications scenarios, we must and have to conduct joint reconstructions for 10 patients/samples altogether no matter in medical or industrial CT imaging.
>
> Q2: As mentioned, "the prior is formulated as a Gaussian distribution in the weight space of the INR", how reliable this distribution can be with only limited samples?
>
> Q3/Q8: Random selecting a few samples to build prior sounds quite empirical and sensitive, without a more systematic study to disclose the reliability of the prior.
>
> Q4: I would respectively disagree with the author that "The INR network architecture itself may provide the ability to mitigate the registration problem", since INR is coordinate-based network modeling. And this algorithm should be quite sensitive to registration issue among prior samples due to the sensitivity and small samples to build the prior.
>
> Q5: The essential difference between MAML and this method is still not very clear, especially regarding learning prior from population samples, even MAML gets prior from more samples. The learn initialization can also be treated as some kind of regularization?
>
> Thus, I would keep my original score as it is. Thanks.

---

> > ### Author Response · Authors · 2023-11-23
> > **Official Response by Authors**
> >
> > Thank you for your continued engagement with our paper and for the additional feedback provided. Upon reviewing your latest comments, we have identified certain areas where there appears to be a misunderstanding about the contributions of our work, as well as some concerns that we believe were addressed in our initial rebuttal. We are grateful for this opportunity to clarify these points further.
> >
> > > Q1: It is still not clear in what kind of applications scenarios, we must and have to conduct joint reconstructions for 10 patients/samples altogether no matter in medical or industrial CT imaging.
> >
> > In our previous response (https://openreview.net/forum?id=vyGp9Mty2t&noteId=7n4cjuJXqC), we outlined several potential scenarios where joint reconstruction could be beneficial in both medical and industrial CT imaging. We wish to emphasize that our approach is not about mandating joint reconstruction in these settings. Instead, it's about showcasing how joint reconstruction can be advantageous, particularly in efficiently utilizing prior information when dealing with sparse data. In practical terms, this could translate into a reduction in the number of scans required for each patient or object, which can potentially lead to benefits in terms of reduced radiation exposure and cost savings.
> >
> > Regarding the specific choice of 10 joint reconstructions, this number was selected to strike a balance between providing enough data to form a meaningful prior and managing computational constraints. Our research isn't suggesting that exactly 10 reconstructions are always optimal; rather, it's a practical decision for baseline experiments. To demonstrate the flexibility and applicability of our method, we included experiments with varying numbers of nodes. These are presented in Figure 4b of the main paper and in Figure 18 of the Appendix (E.8). Figure 4b shows joint reconstruction with different numbers of nodes in an intra-patient scenario, indicating that even a small number of nodes can be sufficient to capture the necessary trends. Conversely, Figure 18 explores the impact of using priors obtained from varying numbers of patients, showing that having more prior nodes can enhance performance when reconstructing unseen patients.
> >
> > > Q2: As mentioned, "the prior is formulated as a Gaussian distribution in the weight space of the INR", how reliable this distribution can be with only limited samples?
> >
> > We value this opportunity to clarify and strengthen the presentation of our research. However, after careful consideration, we believe that our methodology and the decisions made in our study are well-grounded in established research practices.
> >
> > Our choice to use a Gaussian prior in the weight space of the INR is based on a solid foundation of existing research and practices in the field. This approach is **not only theoretically justified but also empirically validated** in various studies (as cited in the paper and our initial response), including our own. To summarize key points,The adoption of a Gaussian distribution for variational families and prior distributions has been a popular approach since 2013 [cited in paper: Kingma & Welling, 2013]; It has been demonstrated that multi-layer perceptrons, similar to INRs, align their weights with a Gaussian distribution post-training [5]. Bayesian neural networks have demonstrated robustness against overfitting in scenarios with limited samples (cited in paper: MacKay, 1992; Neal, 2012; Blundell et al., 2015).
> >
> > Moreover, as detailed in our paper, **our experiments substantiate the effectiveness of our approach, even under the constraints of limited sample sizes.**
> >
> > While we understand and appreciate your concerns regarding the reliability of Gaussian priors with limited samples, the results we have presented clearly indicate their efficacy in the specific context of our study. We believe that these results, combined with the theoretical basis of our approach, adequately address the potential issues associated with limited sample sizes.

---

> > ### Author Response · Authors · 2023-11-23
> > **Official Response by Authors**
> >
> > > Q3/Q8: Random selecting a few samples to build prior sounds quite empirical and sensitive, without a more systematic study to disclose the reliability of the prior.
> >
> > In our experiments, the datasets themselves inherently define the domain of the data. For instance, the LungCT dataset comprises CT scans predominantly centered around the lung area. By randomly selecting from this dataset, we ensure that the learned prior is derived from objects within a similar domain, sharing certain anatomical or structural characteristics. While the LungCT dataset does provide information about the presence of tumors, we deliberately chose not to utilize this specific detail. This decision aligns with the practical reality in medical diagnostics and industrial CT where the presence of certain characteristics is unknown until after the scan is conducted.
> >
> > This approach is practical and relevant because, while the high-level domain (like the type of objects on a production line or the anatomical area in medical CT) is typically predefined, finer sub-domain details (such as specific types of defects or the presence of tumors) are often only discernible post-scan.
> >
> > We believe this method of building priors is **not only sound but also reflects real-world scenarios** where high-level domains are known, but specific sub-domain characteristics are determined after scanning. We are confident that our approach offers a practical solution for efficiently capturing domain-relevant information while maintaining flexibility to adapt to specific characteristics revealed in the scanning process.
> >
> >
> >
> >
> > > Q4: I would respectively disagree with the author that "The INR network architecture itself may provide the ability to mitigate the registration problem", since INR is coordinate-based network modeling. And this algorithm should be quite sensitive to registration issue among prior samples due to the sensitivity and small samples to build the prior.
> >
> >
> >
> >
> > We acknowledge the coordinate-based nature of INR and its inherent sensitivity to spatial coordinates. However, our initial assertion was that the prior learned within the INR's weight space might potentially ease the registration challenge. Our empirical results have shown that our method demonstrates robustness against certain levels of positional deviations. The proof of this robustness stems from our strategy of randomly selecting patients for joint reconstruction, each potentially exhibiting unique positional variations.
> >
> >
> >
> > > Q5:  The essential difference between MAML and this method is still not very clear, especially regarding learning prior from population samples, even MAML gets prior from more samples. The learn initialization can also be treated as some kind of regularization?
> >
> > After careful review, we believe that this question has already been answered in our initial rebuttal in Q5. To ensure clarity, we would like to reemphasize and further elucidate the points made in our initial rebuttal.
> >
> > The learned initialization of MAML can be treated as a regularizer **after** incorporating an manually set early-stop criterion. Only in such cases MAML can be cast as conducting a Maximum A Posteriori (MAP) reconstruction [4]. In this respect, MAML bears some resemblance to our Bayesian framework. However, key distinctions still exist between these two methods. Firstly, our Bayesian framework explicitly sets a Gaussian distribution for the network weights and prior, while the prior of MAML is induced by the early stopping in an implicit manner (so the prior is undetectable). A Gaussian distribution is a natural and effective choice for the weight distribution (as described in the previous question). Secondly, to cast MAML as a Bayesian framework, the integration of individual network weights is assumed to be approximated by making use of a point estimate [4], which introduces the approximation error. Lastly, setting good early stopping criteria is difficult in practice without access to the ground truth image, whereas our Bayesian framework does not require this hyperparameter.

---

> > ### Author Response · Authors · 2023-11-23
> > **Official Response by Authors**
> >
> > Dear Reviewer WCTi
> >
> > Firstly, we would like to express our sincere gratitude for recognizing the strengths of our work in your initial review, particularly the interesting research question we posed and the extensive experiments. Your acknowledgement of these aspects is highly encouraging and underscores the potential impact of our research in the field.
> >
> > We have thoroughly addressed the additional questions raised in your follow-up review, and we hope that our responses have provided clarity and further insight into the quality and depth of our work. Given the detailed explanations and revisions we have provided, we kindly request that you reconsider the score you have initially assigned.
> >
> > We understand and respect your initial assessment and appreciate the critical role it plays in maintaining the high standards of ICLR. However, we believe that our responses to your queries demonstrate a commitment to rigor and thoroughness, qualities that align with the criteria for acceptance.
> >
> > In light of the positive aspects you initially identified and the additional clarifications we have provided, we hope you might find our work has bridged any gaps that were of concern. We believe that our research not only poses an interesting question but also provides substantial evidence and experimentation to support its findings.
> >
> > Thank you for your time and contribution to this review process.

---

### Official Review · Reviewer_ufhh · 2023-11-01

**Soundness:** 2 fair
**Presentation:** 2 fair
**Contribution:** 2 fair
**Rating:** 3
**Confidence:** 4

**Summary:**

The paper deals with the reconstruction of sparse-view CT images. The authors propose a novel Bayesian framework to jointly reconstruct multiple objects using implicit neural representations. The authors evaluate their method against other methods including FBP, iterative, and some joint reconstruction techniques using INRs.

**Strengths:**

The paper is generally well written, and the motivation is clear. Additionally, the literature review is exhaustive. The authors also provide some interesting ablation studies of their method.

**Weaknesses:**

**Writing**

While the paper is well written and easy to read, some statements by the authors are somewhat misleading:
> Sparse-view Computed Tomography (CT) is favored over standard CT for its reduced [...] (Abstract)

suggests, that sparse-view CT would be common practice nowadays, which is -to the best of my knowledge- not the case.

>  While dense measurements typically yield accurate reconstructions, measurements are often intentionally limited to reduce ionizing radiation or cost, resulting in sparse data. (Introduction)

See above, to the best of my knowledge, sparse-view CT is not common in clinical practice. Also, can the authors clarify, how reducing the number of angles may reduce cost?

> While many approaches learn the mapping from sparse-view to dense-view images using supervised learning [...] they often necessitate extensive, domain-specific datasets which are difficult to obtain in practice. (Introduction)

Since full-view acquisitions are the de-facto standard in clinical CT and sparse-view datasets can easily be simulated from these data, such datasets are abundant (e.g., the LDCT Image and Projection data [1] contains over 300 full-view, full-dose acquisitions)

**Experiments**

I have several concerns regarding the experiments:

1. Comparison methods. Unfortunately, the authors do not compare their method against standard CNN-based methods (some of which are also mentioned in the introduction). In particular, the authors do not compare their method against approaches that implicitly (e.g., [2]), or explicitly (e.g., in the form of a DNN) incorporate prior knowledge. The authors also don't compare their method against other, previously proposed INR reconstructing techniques for sparse-view CT.
2. Missing error bars in Fig. 4, 5, & 6. In Tab. 1, over what is the mean $\pm$ standard deviation computed? Are the improvements statistically significant?
3. It is well known, that metrics such as SSIM and PSNR are often not in agreement with quality assessment by clinicians [3,4]. While I recognize that a thorough evaluation involving a reader study is beyond the scope of this work, I don't think the results in Tab. 1 justify the authors claim that their method 'sets a new standard in CT reconstruction performance' (Abstract). Upon visual inspection of Fig. 2 & 3, I find that the proposed method removes many anatomical details and I highly doubt that a clinician would find that reconstructions produced by INR-Bayes are significantly better than those produced by e.g. SingleINR.

**Computational complexity and real-world applicability**

What is the computational complexity of the method? Are all reconstructions performed for the main paper on single $512\times 512$ slices? The application to CBCT reconstructions shown in the appendix is on a much smaller (clinically unusable) image matrix. What would the computational cost for one full patient be compared to the other methods and a CNN-baseline?

[1] https://wiki.cancerimagingarchive.net/pages/viewpage.action?pageId=52758026

[2] Chen GH, Tang J, Leng S (2008) Prior image constrained compressed sensing
(piccs): a method to accurately reconstruct dynamic ct images from highly
undersampled projection data sets. Med Phys 35: 660–663.

[3] Renieblas G P, del Castillo E G, Gómez-Leon N, González A M and Nogués A T 2017 Structural similarity index family for image quality assessment in radiological images J. Med. Imaging.

[4] Verdun F R, Racine D, Ott J G, Tapiovaara M J, Toroi P, Bochud F O, Veldkamp W J, Schegerer A, Bouwman R W, Hernandez-Giron I, Marshall N W and Edyvean S 2015 Image quality in CT: from physical measurements to model observers Phys. Med. 31 823–43

**Questions:**

The experiment configurations intra-patient and 4DCT violate the conditional independence assumption. Does this influence reconstruction quality?

---

> ### Author Response · Authors · 2023-11-17
> **Official Comments by Authors**
>
> Thank you for the thorough review and insightful feedback. We value the time you all have dedicated to reviewing our work. In response, we have carefully prepared a rebuttal and revised the paper accordingly. Below, we address your questions:
>
> > Q1: While the paper is well written and easy to read, some statements by the authors are somewhat misleading.
>
> We are glad to see that the reviewer finds our paper well written and easy to read. We have adjusted our statements in the revision to improve the clarity and the consistency with our empirical findings. The revision should be better aligned with the reviewer’s feedback.
>
> > Q2: “Sparse-view Computed Tomography (CT) is favored over standard CT for its reduced ... (Abstract)” suggests, that sparse-view CT would be common practice nowadays, which is -to the best of my knowledge- not the case.
>
> We adjust the previous statement to “Sparse-view Computed Tomography (CT) has advantages over standard CT for its reduced …”
>
> We realize our phrasing in the abstract may inadvertently suggest that sparse-view CT is widely practiced, which was not our intention. Our goal was to emphasize the potential benefits of sparse-view CT, especially in terms of reduced radiation exposure and increased througput, which are significant considerations in CT. We understand that while sparse-view CT has promising advantages, its widespread adoption is still evolving, often influenced by factors like regulatory approvals and clinical validation.  We note, however, that CT is widely applied to a range of industrial and scientific applications which are not subject to clinical regulatory constraints, and our method may find ready application in these non-clinical domains.
>
> Our work aims to address the critical challenge of image quality in sparse-view CT. By focusing on improving the reconstruction quality from sparse data, we hope to contribute to overcoming one of the key hurdles in its broader implementation. We believe that the current status of sparse-view CT represents a phase in the ongoing advancement and integration of new technologies in practice. Thus, our research seeks to offer solutions that could facilitate its future adoption and underline its relevance in the field.
>
> > Q3: “While dense measurements typically yield accurate reconstructions, measurements are often intentionally limited to reduce ionizing radiation or cost, resulting in sparse data. (Introduction)”
> >
> > See above, to the best of my knowledge, sparse-view CT is not common in clinical practice. Also, can the authors clarify, how reducing the number of angles may reduce cost?
>
> Thank you for highlighting this aspect. We have revised our introduction to better articulate the context and potential advantages of sparse-view CT. Our new wording in the introduction is: “In specific situations, limiting the number of CT measurements can offer benefits such as reduced radiation exposure and cost management, which may lead to the use of sparse data.” This refined statement more accurately reflects the targeted use of sparse-view CT, emphasizing its potential benefits without suggesting widespread current clinical adoption.
> In the realm of medical CT imaging, reducing the number of scanning angles can decrease patient exposure to ionizing radiation, a crucial consideration in frequent or high-risk scanning scenarios. This approach can also expedite the scanning process, potentially reducing wait times and increasing patient throughput in busy clinical settings. It can also potentially reduce the cost by lowering operational costs associated with extended scanner usage.
> In industrial applications, where CT scanners often operate continuously in production environments, reducing scanning angles can significantly enhance efficiency. Shorter scan times increase throughput and reduce operational delays, directly impacting productivity. Additionally, decreased scanner usage results in lower energy requirements, contributing to cost savings and reduced environmental impact.
> We believe that emphasizing these specific contexts provides a clearer understanding of where sparse-view CT can offer substantial advantages in both medical and industrial settings. We appreciate your feedback and hope this revised explanation addresses your concerns.

---

> ### Author Response · Authors · 2023-11-17
> **Official Comments by Authors**
>
> > Q4: “While many approaches learn the mapping from sparse-view to dense-view images using supervised learning [...] they often necessitate extensive, domain-specific datasets which are difficult to obtain in practice. (Introduction)”
> >
> >Since full-view acquisitions are the de-facto standard in clinical CT and sparse-view datasets can easily be simulated from these data, such datasets are abundant (e.g., the LDCT Image and Projection data [1] contains over 300 full-view, full-dose acquisitions)
>
> While it is true that full-view acquisitions are standard in clinical CT and sparse-view datasets can be easily simulated from these data, the challenge lies not merely in dataset availability. The primary difficulty is in obtaining domain-specific datasets that are sufficiently diverse and representative to train models effectively for a wide range of real-world applications. While simulated sparse-view datasets derived from full-view acquisitions provide a valuable resource, they may not fully capture the complexity and variability found in actual sparse-view scenarios. Secondly, in industrial contexts, new products frequently lack comprehensive full-view acquisitions. Our research focuses on methods that can generalize well from limited or sparse data. By using methods that do not solely rely on large, domain-specific datasets, we aim to improve the robustness and applicability of our models in diverse real-world scenarios, where obtaining such datasets might be impractical.

---

> ### Author Response · Authors · 2023-11-17
> **Official Comments by Authors**
>
> > Q5: Comparison methods. Unfortunately, the authors do not compare their method against standard CNN-based methods (some of which are also mentioned in the introduction). In particular, the authors do not compare their method against approaches that implicitly (e.g., [2]), or explicitly (e.g., in the form of a DNN) incorporate prior knowledge. The authors also don't compare their method against other, previously proposed INR reconstructing techniques for sparse-view CT.
>
> Our study focuses on approaches that excel in sparse-data environments. We selected comparison methods aligned with this focus, aiming to demonstrate the effectiveness of our approach in scenarios where data is inherently limited or sparse. This includes avoiding comparisons with methods that rely heavily on full-view acquisitions, as these do not align with the primary objective of our research, which is to excel in limited data scenarios. We have added sentences to the introduction section for clearer elaboration: “There are also works that adopt heuristic image priors, e.g., Total Variation (TV) [4,5,6], or use dense view images as priors [2, 7] to assist in reconstruction. However, these methods often lack domain-specific enhancements or require information from dense-view images.”
>
> We acknowledge that our study did not include comparisons with standard CNN-based methods or approaches that implicitly or explicitly incorporate prior knowledge, such as the ones mentioned in [2]. The primary focus of our research was to explore the potential of Implicit Neural Representations (INRs) in utilizing the statistical regularities shared among different objects with similar representations to enhance CT reconstruction quality through joint reconstruction. Therefore, we primarily concentrated on comparing our method with other prior-embedding INR-based methods that align closely with our research question as stated in the introduction. We have revised the last sentence of the introduction to better reflect the scope of our work and the research question posed: “Our results establish that our method either outperforms or is competitive with existing INR-based baselines.” This aligns with our initial inquiry: “Can INRs utilize the statistical regularities shared among different objects with similar representations to enhance reconstruction quality through joint reconstruction?”
>
> Regarding [2], it proposes an intriguing method of incorporating prior information from a previous image within a compressed sensing framework for 4DCT. While this approach is practically feasible for 4DCT, where scanning angles of different phases can be interleaved to form a dense-view reconstruction, it does not necessarily translate to our intra-patient and inter-patient settings. In these scenarios, even if the scans are conducted with interleaved angles, the substantial differences between the images can make it challenging to construct a cohesive and informative dense-view prior. This difference in the nature of the data and the specificities of our research focus led us to prioritize comparisons within the scope of INR-based methods.
>
> Our study compares our method with a diverse range of techniques, including classical methods like FBP and SIRT, as well as advanced INR-based approaches like Single INR [8][9], FedAvg[11], MAML[10], and INRWild[12]. We also add a new baseline Nerp in Appendix F.7 in the revision. Our selection of comparison methods was driven by the desire to demonstrate the effectiveness of our approach across a spectrum of techniques, from classical to the latest INR-based methods. We thank the reviewer for also acknowledging our exhaustive literature review, as we believe this diverse comparison set provides a comprehensive assessment of our method's capabilities in handling sparse-view CT reconstruction challenges. If the reviewer thinks there is any relevant INR for sparse-view CT that we have missed, we would be very happy to receive this feedback for incorporation during the discussion period or later. Comparing our method against a broader range of techniques, including other prior incorporation methods, are interesting for us as future work.

---

> ### Author Response · Authors · 2023-11-17
> **Official Comments by Authors**
>
> > Q6: Over what is the mean standard deviation computed?
>
> The mean and standard error reported in our study are computed over all reconstructed images of our experiments, taking into account both random initializations and variations in image selections.
>
> For instance, in the case of the 4DCT dataset, which has dimensions of 10x136x512x512, we perform joint reconstructions using groups of 10 images, each being 512x512 in size. To ensure a diverse set of data, we randomly select these images from indices ranging between 20 and 120 within the dataset. This approach yields a substantial sample size for each experiment, specifically, we calculate the mean and standard error across 1,000 images (10 images per group across 100 groups).
>
> We appreciate your feedback as it highlighted the need for clearer explanation in our manuscript. To address this, we added the sentence “We calulate mean and standard error over all reconstructioned images in each experiment.” in the metrics subsection of section 5 (Experiments), offering a more detailed account of how these statistical measures were derived.
>
> > Q7: Are the improvements statistically significant?
>
> To rigorously evaluate the statistical significance of our method's improvements, we conducted a Wilcoxon signed-rank test comparing our results with those of the second-best method. The obtained p-values, being less than 0.05 in most cases, strongly suggest that these improvements are indeed statistically significant. For clarity and transparency, we present the detailed Wilcoxon signed-rank test results below:
>
> | Experiment | Sample Size | Second Best Method | W-statistic (PSNR) | P-value (PSNR) | W-statistic (SSIM) | P-value (SSIM) |
> |------------|-------------|--------------------|--------------------|----------------|--------------------|----------------|
> | Intra-patient | 950 | MAML | 13252 | 2.16e-139  | 1617 | 7.72e-155 |
> | Inter-patient Lung | 100 | MAML | 348 | 7.14e-14 | 67 | 2.88e-17 |
> | Inter-patient Brain| 50 | MAML | 614 | 0.83 | 597 | 0.70 |
> | 4DCT | 1000 | FedAvg | 186785 | 3.73e-12 | 4689 | 3.79e-159 |
> | Adapt to new patient | 50 | MAML | 21 | 7.94e-13 | 0 | 1.78e-15 |
>
> We have now highlighted in bold the results of both MAML and our method in Table 1 row 5 (Inter-patient Brain)  to demonstrate their statistically significant improvement over other methods. We appreciate your valuable feedback and hope that this additional analysis provides the necessary clarification.
>
>
> > Q8: It is well known that metrics such as SSIM and PSNR are often not in agreement with quality assessment by clinicians [3,4]. While I recognize that a thorough evaluation involving a reader study is beyond the scope of this work, I don't think the results in Tab. 1 justify the authors claim that their method 'sets a new standard in CT reconstruction performance' (Abstract). Upon visual inspection of Fig. 2 & 3, I find that the proposed method removes many anatomical details and I highly doubt that a clinician would find that reconstructions produced by INR-Bayes are significantly better than those produced by e.g. SingleINR.
>
> We concur that PSNR and SSIM metrics may not always align with practical clinical assessments and acknowledge that incorporating human evaluation falls outside the scope of this paper and the ICLR conference. In evaluating our framework, it was necessary to select certain metrics for numerical comparison with other methods. Therefore, we chose PSNR and SSIM, which remain widely accepted and standard in both the machine learning and medical imaging fields, as evidenced by references [13], [14], [15], [16]. While we recognize that these metrics are not flawless, we believe that their limitations should not be viewed as a fundamental weakness of our paper.
>
> To address your concerns, we have introduced a new section on limitations in our conclusion, highlighting the potential discrepancies between numerical metrics and clinical evaluations. The new limitation section now is: "We recognize that the metrics employed in our study may not always correlate with clinical evaluations [3][4]. If applied in a medical application, clinical verification of our method remains essential to understand its practical implications and efficacy in a given clinical setting." This addition aims to clarify the context and application of our findings.
>
> Additionally, we have carefully revised the abstract to more accurately represent our study's results and scope. The updated abstract now is: "Our results indicate marked advancements over baseline methods in standard numerical metrics, marking a progressive stride in CT reconstruction techniques." This revision aligns our claims with the evidence presented, while acknowledging the nuances of clinical applicability.

---

> ### Author Response · Authors · 2023-11-17
> **Official Comments by Authors**
>
> > Q9: What is the computational complexity of the method?
>
> We understand the importance of this aspect and have added a new section in the appendix that details the computation costs. For your convenience, we have replicated the information in Table 5 [here](https://openreview.net/forum?id=vyGp9Mty2t&noteId=CI64s0jmxF), which presents a comparative analysis of the computational requirements.
>
> Due to the increased model capacity and therefore higher optimization complexity, our method exhibits slightly higher computational costs compared to other methods, but the increase is relatively small (less than 10%).
>
> > Q10: Are all reconstructions performed for the main paper on single 512 \times 512 slices? The application to CBCT reconstructions shown in the appendix is on a much smaller (clinically unusable) image matrix. What would the computational cost for one full patient be compared to the other methods and a CNN-baseline?
>
> Regarding the scope of our experiments, the main paper focuses primarily on slice-based analyses. For the 3D Cone-Beam Computed Tomography (CBCT) experiments presented in the appendix, we utilize volumes of size 128^3. This 3D experiment was designed as a proof-of-concept to demonstrate our method's applicability and effectiveness in a 3D context.
>
> We acknowledge that the 128^3 volume size used in the 3D CBCT experiment is smaller than what is typically used in clinical settings. The choice of this size was primarily driven by computational constraints. For a comprehensive comparison, our experiments iterate over the whole dataset, which involves running hundreds of reconstructions across various methods. The total computational load is substantial. Thus, to manage these constraints effectively while still providing meaningful insights, we opted for smaller volume sizes for the 3D experiments.
>
> To provide a clearer perspective on the computational aspects, we included a new section in the Appendix (E6) that compares the computational costs across various INR-based methods, including our own. We also copied the comparison table [here](https://openreview.net/forum?id=vyGp9Mty2t&noteId=CI64s0jmxF) for your easier reference. While this comparison uses slice-based experiments, it offers a framework to extrapolate potential computational costs for larger, clinically relevant volumes.
>
> We hope this information clarifies your queries regarding the computational complexity and the scope of our experiments. We are continually working towards optimizing the method to handle larger volumes more efficiently.
>
> > Q11: The experiment configurations intra-patient and 4DCT violate the conditional independence assumption. Does this influence reconstruction quality?
>
> The conditional independence assumption allows us to decompose the variational inference problem related to the joint posterior distribution across objects. This assumption significantly simplifies the model and is instrumental in deriving an efficient algorithm for our Bayesian framework. Our experiments have shown that the INR-Bayes framework performs robustly in 4DCT and intra-patient configurations, producing high-quality reconstructions. This indicates that the framework’s capability to learn complex representations effectively compensates for any limitations introduced by the conditional independence assumption.

---

> ### Author Response · Authors · 2023-11-17
> **Official Comments by Authors**
>
> We appreciate your insightful feedback and have diligently addressed the concerns raised. We hope that our revisions and clarifications demonstrate the robustness and relevance of our work, and we respectfully invite you to re-evaluate our submission in light of these updates.

---

> ### Author Response · Authors · 2023-11-17
> **Official Comments by Authors**
>
> [4] Sidky and Pan, Image reconstruction in circular cone-beam computed tomography by constrained, total-variation minimization, Physics in Medicine & Biology, 2008
>
> [5] Liu et. al, Total variation-stokes strategy for sparse-view x-ray ct image reconstruction, Trans. Medical Imaging, 2013
>
> [6] Zang et al, Super-resolution and sparse view ct reconstruction, ECCV 2018
>
> [7] Shen et. al, Nerp: implicit neural representation learning with prior embedding for sparsely sampled image reconstruction. Trans Neural Networks and Learning Systems, 2022.
>
> [8] Zha et. al, Intratomo: self- supervised learning-based tomography via sinogram synthesis and prediction, CVPR 2021
>
> [9] Zhang et. al, Naf: Neural attenuation fields for sparse-view cbct reconstruction, MICCAI 2022
>
> [10] Tanick et. al, Learned initializations for optimizing coordinate-based neural representations, CVPR 2021
>
> [11] Kundu et. al, Panoptic neural fields: A seman- tic object-aware neural scene representation, CVPR 2022
>
> [12] Martin-Brualla et. al, Nerf in the wild: Neural radiance fields for unconstrained photo collections, CVPR 2021
>
> [13] Song et. al, Solving Inverse Problems in Medical Imaging with Score-Based Generative Models, ICLR 2023
>
> [14] Wu et. al, Unsupervised Polychromatic Neural Representation for CT Metal Artifact Reduction, NIPS 2023
>
> [15] Zhou et. al, dudoufnet:  Dual-Domain  Under-to-Fully-Complete  Progressive  Restoration  Network  for  Simultaneous  Metal  Artifact  Reduction  and  Low-Dose  CT  Reconstruction, Trans. Medical Imaging, 2022
>
> [16] Hu et. al, dior:   Deep   Iterative   Optimization-Based   Residual-Learning   for   Limited-Angle   CT   Reconstruction, Trans. Medical Imaging, 2022

---

> ### Author Response · Authors · 2023-11-21
> **Rebuttal follow up**
>
> As we are now approaching the end of the reviewer-author discussion
> period, we would like to follow up on our rebuttal response and gently
> remind you to provide your valuable feedback.
>
> We are keen to know if our response and improvements address your
> concerns and satisfy you. Furthermore, we would like to confirm if there
> are any further concerns or questions you may have regarding our work,
> we are more than willing to engage in a productive discussion during the
> reviewer-author period.
>
> Thank you for your valuable input and time.

---

> > ### Comment · Reviewer_ufhh · 2023-11-21
> >
> > I would like to thank the authors for providing such a detailed rebuttal which addresses some of my concerns. I still have some comments and questions:
> >
> > I still think that the authors use misleading writing also in the new paragraphs:
> > > [...] This approach can also expedite the scanning process, potentially reducing wait times and increasing patient throughput in busy clinical settings. It can also potentially reduce the cost by lowering operational costs associated with extended scanner usage. In industrial applications, where CT scanners often operate continuously in production environments, reducing scanning angles can significantly enhance efficiency. Shorter scan times increase throughput and reduce operational delays, directly impacting productivity.
> >
> > Can the authors comment on why scanning from fewer angles reduces acquisition times in practice? The CT spins continously at over 100 rpm. The amount of projections acquired should not influence scan time at all (at least for clinical CT, for nondestructive testing this could make a difference). The only factor that can really reduce cost is reconstruction time. Is this faster for the authors method compared to an FBP from the full set of projections? Am I correct in assuming that the numbers in the newly added Table 5 refer to reconstruction of 10 images @ $512\times 512$ each? If this really takes 10 hours with the proposed method it clearly is impractical and would in fact substantially increase cost. It would mean that reconstructing e.g. chest CTs (usually having about 200 slices each) from 10 patients would require approximately 83 days... Can the authors comment on the computation time of the iterative reconstruction? It should be on the order of few hours for the entire 3D scan and thus 2 orders of magnitude faster. And iterative reconstruction is considered to be too slow for many clinical settings (e.g. for use in ER). If the proposed method really is this slow, this needs to be mentioned as a **major** limitation in the discussion.
> >
> > > This less than 10% increase in time is attributed to the added model capacibility and the Gaussian noise sampling procedure in INR-Bayes. However, considering the enhanced reconstruction quality and robustness achieved, this additional time investment can be justified.
> >
> > In light of the point above, I find this newly added statement in the appendix highly misleading. A reader could think that this is an adequate reconstruction time, when in fact it is not. Why didn't the authors add times for FBP and the iterative (SART, I don't know why the authors call it SIRT instead) algorithm? FBP should take < 1s and iterative <1min for a single slice, so several orders of magnitude faster than the proposed (and all INR) method.
> >
> > > While simulated sparse-view datasets derived from full-view acquisitions provide a valuable resource, they may not fully capture the complexity and variability found in actual sparse-view scenarios
> >
> > I don't quite understand what the authors mean with this point. Only using every $n^{\text{th}}$ projection models the system perfectly.
> >
> > > We acknowledge that the 128^3 volume size used in the 3D CBCT experiment is smaller than what is typically used in clinical settings. The choice of this size was primarily driven by computational constraints. For a comprehensive comparison, our experiments iterate over the whole dataset, which involves running hundreds of reconstructions across various methods. The total computational load is substantial. Thus, to manage these constraints effectively while still providing meaningful insights, we opted for smaller volume sizes for the 3D experiments.
> >
> > I would like to thank the authors for this clarification. How many datasets were included in the 3D CBCT experiments (unfortunately, I couldn't find this information in the appendix)? I still don't think that evaluating SSIM or PSNR on such small volumes makes much sense since it is extremely far from clinical practice.

---

> > > ### Author Response · Authors · 2023-11-22
> > > **Official Response by Authors**
> > >
> > > We sincerely appreciate your active engagement during the discussion period and the time you have dedicated to providing concrete suggestions. Your insights have allowed us to further clarify and refine key aspects of our work. In the following sections, we address each of your remaining questions in detail.
> > >
> > > > Q1:  Why scanning from fewer angles reduces acquisition times in practice?
> > >
> > > In CT imaging, machines can offer two distinct scan modes: sequential (stop-and-go during scanning) and continuous (non-stop during scanning). Although detailed user instructions for CT machines are not widely available online, we found supporting documentation (under CCT Mode - CCT Single or CCT Continuous, page 5-32) for the 'Philips Incisive CT' series [17] and the product description (under Fast scanning with VAST mode) of the Zeiss 'METROTOM' series [18], which illustrate these modes. In the sequential scanning mode, the reduction of scanning angles can lead to shorter scanning times.
> > >
> > > We acknowledge that standard clinical settings usually employ continuous scan modes, reducing the number of projections does not necessarily shorten the scan time. However, a variety of more customized clinical systems are currently brought to market, such as specialized breast-CT systems [https://ab-ct.com/] and devices for scanning fractures [https://www.planmed.com/], that utilize sequential acquisition techniques.
> > >
> > > Therefore, in our introduction, we cautiously state that 'In specific situations, limiting the number of CT measurements can offer benefits such as reduced radiation exposure and cost management, potentially leading to the use of sparse data.' This statement aims to reflect the varied and evolving nature of CT scanning practices, including those beyond standard clinical settings.
> > >
> > >
> > > > Q2: The only factor that can really reduce cost is reconstruction time.
> > >
> > > We kindly disagree with this strong argument. Reconstruction time is an important factor of cost, and we have discussed in the previous answer that this issue of INRs is expected to be addressed. There are many other factors that have an impact on the cost. Sparse-view CT can potentially reduce costs by decreasing scanner utilization. Even in continuous scanning modes, acquiring fewer projections can lead to less use of the X-ray source and detectors. This reduced usage can translate to lower operational costs, as it may lessen wear and tear on equipment and reduce energy consumption. As you also mentioned, in nondestructive testing scenarios, reducing the number of projections can contribute to shorter scan times and potentially lower costs. This aligns with our statement in the introduction that 'In **specific situations**, limiting the number of CT measurements can offer benefits such as **reduced radiation exposure** and **cost management**, potentially leading to the use of sparse data.' We believe this statement accurately reflects the multifaceted nature of cost considerations in CT imaging.

---

> > > ### Author Response · Authors · 2023-11-22
> > > **Official Response by Authors**
> > >
> > > > Q3: Regarding computation time and practical application of our method:
> > >
> > > We acknowledge that computation complexity is an important issue, however we kindly point out that it shouldn’t become the only concern that blocks the study of other issues. While our method and other INR-based methods currently face challenges with computation time, it excels in providing superior reconstruction quality. Our research is particularly focused on whether INRs can utilize statistical regularities shared across various objects to enhance reconstruction quality (as stated in the beginning of the paper). Enhancing the overall efficiency of INRs is orthogonal to the focus of this study, but we note that research in this area is active and progressing fast. For example, [22] achieves 1000 iterations for a bit more than 1 second on an image regression task (demo [https://github.com/NVlabs/tiny-cuda-nn]). We are optimistic about addressing the relatively long computational time of INRs in the future. Additionally, our results of the computation time is measured on a middle-low end device. Computational hardware is developing rapidly [https://www.nvidia.com/en-us/geforce/graphics-cards/compare/], indicating a promising future where such computational challenges become manageable.
> > >
> > > Furthermore, the application of INRs in CT reconstruction is a relatively new venture, and our research aims to deepen the understanding of INRs in this domain. The joint reconstruction topic is relevant to this field, and our proposed Bayesian framework is novel. We also conducted extensive experiments to compare our methods with baselines. Our contribution to the community is clear according to the ICLR criterion [https://iclr.cc/Conferences/2023/ReviewerGuide].
> > >
> > > To fully address concerns regarding computational efficiency, we have adopted your suggestion and amended the limitations section in our discussion to include: “We recognize that INR-based methods outperform conventional ones but require more computation, making their efficiency a crucial focus for future research”.
> > >
> > >
> > >
> > > > Q4: Confusion about the statement “This less than 10% increase in time…”. Why didn’t compare with FBP and SIRT?
> > >
> > >
> > > We appreciate your feedback and understand the concern about the potential misinterpretation of our computational efficiency statement. Our comparison in Table 5 is specifically among INR-based methods, as these are generally more computationally demanding due to their reliance on neural networks. We agree that FBP and SIRT are significantly faster, with FBP typically taking less than a second and SIRT less than a minute on GPU. However, these methods also typically yield lower reconstruction performance compared to INR-based methods, as demonstrated in our experiments.
> > >
> > > To avoid any misinterpretation, we have revised our statement in the appendix to specify that the less than 10% increase in computational time refers only to a comparison with other INR-based methods (SingleINR, FedAvg, and MAML). Additionally, to provide a more comprehensive perspective, we have included the computation times for FBP and SIRT in the caption of Table 5. This change should help clarify that the additional time investment for INR-Bayes is in the context of similar INR-based approaches and not in comparison to the much faster FBP and SIRT methods.
> > >
> > >
> > > > Q5: The iterative (SART, I don't know why the authors call it SIRT instead) algorithm?
> > >
> > > We appreciate the opportunity to clarify the distinction between SIRT and SART. Indeed, in some contexts, particularly within the mathematics community, SIRT is sometimes referred to as SART [19]. However, it is important to note that while SIRT and SART are both iterative reconstruction methods with similarities, they are distinct techniques [20, 21]. For instance, the Tigre toolbox [21] provides separate implementations for both SIRT and SART. In our paper, we have specifically referred to and compared our method with SIRT, not SART.

---

> > > ### Author Response · Authors · 2023-11-22
> > > **Official Response by Authors**
> > >
> > > > Q6: Only using every n-th projection models the system perfectly.
> > >
> > > We are glad to further clarify our points. Using every n-th projection to create a sparse dataset is indeed a valid approach for simulating sparse-view CT scans, and it effectively models the physical process of acquiring fewer projections. Our previous comment as in [here](https://openreview.net/forum?id=vyGp9Mty2t&noteId=VvOYhP4CzT) was aimed at highlighting the potential limitations of this method when applied to diverse scenarios.
> > >
> > > For instance, networks trained on sparse-view data derived from full-view acquisitions of a specific patient group (e.g., patients with tumors) might not generalize effectively to other patient groups (e.g., patients without tumors). This potential limitation stems from the fact that such networks learn a mapping that is inherently tied to the characteristics of the training data.
> > >
> > > The performances of those methods highly depend on the diversity and availability of dense-view information. Our method, by contrast, is designed to reconstruct images directly from sparse measurements without relying on matched dense-view information. This approach offers flexibility and adaptability across different scanning scenarios, as it is not constrained by the need for corresponding dense-view data.
> > >
> > >
> > > > Q7: How many datasets were included in the 3D CBCT experiments (unfortunately, I couldn't find this information in the appendix)? I still don't think that evaluating SSIM or PSNR on such small volumes makes much sense since it is extremely far from clinical practice.
> > >
> > > In the 3D CBCT experiments detailed in Table 3, we evaluated our method against FBP, SIRT, SingleINR, FedAvg, and MAML across 9 different groups. Each group performed joint reconstruction on 10 different patients' CT volumes of size 128^3, amounting to a total of 90 patient volumes. For clarity, we have included this detail in Section E.1 of the appendix: 'Experiments involved 9 groups of joint reconstructions, each reconstructing 10 different patients' CT volumes sized at 128^3.
> > >
> > > As we have previously noted, INR is an emerging area in the field of CT reconstruction, and to our knowledge, no INR-based method has yet been adopted in clinical practice. Our study aims to contribute to the advancement of this field, rather than to produce an immediate industrial or clinical solution. By demonstrating the capabilities of our proposed Bayesian method in a proof-of-concept manner, we aim to illustrate its potential utility in furthering CT reconstruction research.

---

> > > ### Author Response · Authors · 2023-11-22
> > > **Official Response by Authors**
> > >
> > > ---
> > > Once again, we greatly appreciate your continued engagement and insightful questions. We have endeavored to address these additional concerns with the same thoroughness and attention to detail as in our initial rebuttal. We believe that the clarifications and supplementary information provided in this response further strengthen our paper’s contribution to sparse-view CT reconstruction using INRs.
> > >
> > > Given the time-sensitive nature of this phase, we are prepared to promptly address any further concerns you might have, to facilitate a re-evaluation before the deadline. Your feedback has been invaluable, and a revised evaluation, should you deem it appropriate, could significantly impact the overall assessment of our work. We are hopeful that our responses meet your expectations for a higher score.
> > >
> > > We remain at your disposal for any last-minute clarifications and look forward to your final evaluation. Thank you once again for your dedication and valuable insights.
> > >
> > > ---
> > >
> > > [17] https://www.documents.philips.com/assets/Technical%20Reference%20Guide/20220406/ac1549b201354dd2a2beae6f00794fdd.pdf?feed=ifu_docs_feed
> > >
> > > [18]
> > > https://www.zeiss.com/metrology/products/systems/computed-tomography/scattercontrol.html
> > >
> > > [19] Hansen et. al, Computed Tomography, SIAM, 2021, Chapter Algebraic Iterative Reconstruction Methods, page 219-220
> > >
> > > [20] Gregor and Fessler, Comparison of SIRT and SQS for Regularized Weighted Least Squares Image Reconstruction, Trans. Comp. Imaging, 2015
> > >
> > > [21] Biguri et. al, TIGRE: a MATLAB-GPU toolbox for CBCT image reconstruction, Biomed. Phys. Eng., 2016
> > >
> > > [22] Mueller et. al, Instant Neural Graphics Primitives with a Multiresolution Hash Encoding, ACM Trans. Graph., 2022

---

> > > ### Author Response · Authors · 2023-11-23
> > > **Rebuttal follow up**
> > >
> > > Dear Reviewer ufhh,
> > >
> > > As the discussion phase of ICLR is drawing to a close, we wanted to extend our appreciation for your valuable insights and feedback throughout this process. Your initial concerns have been instrumental in guiding the improvements we've made to our paper. We have diligently addressed each point you raised in our responses and have made corresponding revisions to our submission.
> > >
> > > We understand that the review process is demanding and time-sensitive, but if you find that we have satisfactorily addressed your concerns, we would be grateful if you could reconsider your evaluation and score of our work.
> > >
> > > Your reevaluation is crucial for us, and we hope our efforts to address your feedback reflect our commitment to advancing the quality and impact of our research. Thank you once again for your time and thoughtful consideration.

---

### Author Response · Authors · 2023-11-17
**General Response to Reviewer Feedback**

We thank all reviewers for their constructive and valuable feedback and greatly appreciate the time spent reviewing our work. Accordingly, we have carefully prepared our rebuttal and revised the paper based on your insights.

In the revision, all modified text has been marked in red to facilitate easy identification of changes. Additionally, we have added new experiments and a schematic illustration, details of which are referenced in the individual response.

---

### Meta-Review · Area_Chair_y68V · 2023-12-13

**Metareview:**

The authors propose an implicit neural representation approach for the sparse CT reconstruction problem, which utilizes population measurements to learn an image prior and improve individual reconstructions. This was a borderline paper with reviewers expressing concerns about the clinical relevance of the proposed sparse CT and population prior settings. The reviewers agreed that the method was novel, clearly explained, and well evaluated but there are weaknesses.

**Justification For Why Not Higher Score:**

Concerns regarding the clinical/practical relevance of the proposed sparse view CT setting, and the population reconstruction approach.

**Justification For Why Not Lower Score:**

N/A

---

### Decision · Program_Chairs · 2024-01-16

Reject